# VPS34 K29/K48 branched ubiquitination governed by UBE3C and TRABID regulates autophagy, proteostasis and liver metabolism

Yu-Hsuan Chen[1,2], Tzu-Yu Huang[1,2], Yu-Tung Lin[1,2], Shu-Yu Lin[1], Wen-Hsin Li[1], Hsiang-Jung Hsiao[1,3], Ruei-Liang Yan[1,3], Hong-Wen Tang[1], Zhao-Qing Shen[4], Guang-Chao Chen [1,2], Kuen-Phon Wu [1,2], Ting-Fen Tsai[4] & Ruey-Hwa Chen [1,2,3 ✉]

The ubiquitin–proteasome system (UPS) and autophagy are two major quality control processes whose impairment is linked to a wide variety of diseases. The coordination between UPS and autophagy remains incompletely understood. Here, we show that ubiquitin ligase UBE3C and deubiquitinating enzyme TRABID reciprocally regulate K29/K48-branched ubiquitination of VPS34. We find that this ubiquitination enhances the binding of VPS34 to proteasomes for degradation, thereby suppressing autophagosome formation and maturation. Under ER and proteotoxic stresses, UBE3C recruitment to phagophores is compromised with a concomitant increase of its association with proteasomes. This switch attenuates the action of UBE3C on VPS34, thereby elevating autophagy activity to facilitate proteostasis, ER quality control and cell survival. Specifically in the liver, we show that TRABID-mediated VPS34 stabilization is critical for lipid metabolism and is downregulated during the pathogenesis of steatosis. This study identifies a ubiquitination type on VPS34 and elucidates its cellular fate and physiological functions in proteostasis and liver metabolism.

[1] Institute of Biological Chemistry, Academia Sinica, Taipei, Taiwan. [2] Institute of Biochemical Sciences, College of Life Science, National Taiwan University, Taipei, Taiwan. [3] Institute of Molecular Medicine, College of Medicine, National Taiwan University, Taipei, Taiwan. [4] Department of Life Sciences and Institute of Genome Sciences, National Yang-Ming University, Taipei, Taiwan. ✉email: rhchen@gate.sinica.edu.tw

Protein ubiquitination, a major post-translational mechanism, is mediated by the consecutive actions of E1, E2, and E3 enzymes and is antagonized by deubiquitinating enzymes (DUBs)[1,2]. Since ubiquitin contains seven lysine residues and one N-terminal methionine residue for linking to another ubiquitin moiety, polyubiquitin chain can adopt highly diverse topologies to result in distinct functional outcomes[3,4]. In the homotypic chain, all ubiquitin moieties are linked through the same lysine or methionine residue and a total of eight different chain types can be formed. Additionally, heterotypic chain containing more than one linkage types can also be formed and is further categorized into linear (mixed) and branched chains[4].

Compared to the homotypic chains, characterization of branched polyubiquitin chain is technically challenging. However, the recently developed Ub-clipping method has greatly advanced the detection and quantification of branched ubiquitin chains[5]. This led to the finding that 10–20% of the cellular polyubiquitin chains exist as the branched chains. However, only few types of branched polyubiquitin chains have been identified and characterized[6,7]. The K11/K48-branched chain was discovered on APC/C substrates, ERAD substrates, and cytoplasmic misfolded proteins[8–10]. Importantly, formation of this chain type not only is highly induced by proteotoxic stresses, but also leads to an enhanced substrate degradation by proteasome[4,9], indicating its critical role in the protein quality control. The K48/K63 branched ubiquitin chain, which is often generated by a sequential action of the K63- and K48-specific E3 ligases, is induced in response to interleukin-1β. Formation of this branched chain either prevents deubiquitination by K63-specific DUB to amplify the signaling effect of K63 ubiquitination or switches the fate of substrates towards proteasomal degradation[11,12]. The K29/K48 heterotypic ubiquitin chain was identified by a pull-down experiment using a ubiquitin-binding domain (UBD) specific for K29-linked ubiquitin chain[13]. This study revealed that most, if not all, cellular K29 linkages exist within the heterotypic chains that also contain K48 linkages. A subsequent middle-down mass spectrometry analysis identified the branched nature of this heterotypic chain[14]. Given that K29 linkage is the most abundant atypical linkage in mammalian cells, representing 8–9% of the total cellular ubiquitin linkages[5,15], it is expected that a substantial amount of cellular proteins is conjugated by this chain type. However, protein being modified by this branched chain has only been discovered from the model substrate of ubiquitin-fusion degradation (UFD) pathway in yeast[16]. Formation of this branched chain promotes proteasomal degradation of the modified proteins.

Besides the ubiquitin–proteasome system (UPS), autophagy is another major quality control pathway, which involves the formation of double-membrane autophagosome for lysosomal degradation of cytosolic components[17,18]. UPS and autophagy are both critical for the elimination of misfolded, unfolded, and damaged proteins[19]. In general, soluble and monomeric proteins are preferentially targeted by UPS, whereas their insoluble aggregates or oligomeric form are removed by autophagy[20]. Failure of either process disturbs cellular proteostasis and has been associated with aging and a large class of age-related proteinopathies, including neurodegeneration, metabolic disorders, and cancer, among others[21]. Although the two processes are operated through independent molecular mechanisms, recent studies have revealed their interconnections at multiple points to facilitate a mutually regulated or concerted action[22–26]. However, it remains poorly understood whether and how misfolded/unfolded proteins themselves can trigger a mechanism to coordinate the activities of these two processes. Identification of such mechanism may benefit the development of therapeutic strategies for treating various proteinopathies.

One type of the interplays between UPS and autophagy is the ubiquitin-dependent regulation of autophagic proteins[27]. Among the major components of autophagy pathway, the class III PI3-kinase complex (also known as the VPS34 complex) is the hub of ubiquitin-dependent regulation, along with other types of post-translational modifications[28,29]. This complex catalyzes the production of PI3P and is required for both bulk and selective types of autophagy by controlling both autophagosome formation and maturation[30–32]. Therefore, it is conceivable that multiple signals converge on this complex for regulating autophagy. Accordingly, the complex subunit Beclin-1 is a target of multiple E3 ligases and DUBs[27]. However, ubiquitin-mediated regulation of the catalytic subunit VPS34 is less reported. The FBXL20-based Cullin 1 complex promotes VPS34 ubiquitination and degradation in response to DNA damage[33], whereas KLHL20-based Cullin 3 complex targets VPS34 for ubiquitination and degradation to contribute to autophagy termination[34]. A recent study identified a role of USP13 in VPS34 deubiquitination and stabilization and NEDD4-1 autoubiquitination promotes the recruitment of USP13 to VPS34[35]. In addition, VPS34 K63-linked polyubiquitination in *C. elegans* mediated by E2 enzyme UBC-13/UEV-1 and E3 enzyme CHN-1 leads to VPS34 stabilization[36], consistent with a nondegradable fate of this chain type[3].

In this study, we report that VPS34 is regulated by K29/K48-branched ubiquitination through the reciprocal actions of ubiquitin ligase UBE3C and DUB TRABID. This ubiquitination enhances VPS34 binding to proteasome for degradation, leading to autophagy inhibition. Under basal and starvation, UBE3C and TRABID coordinately regulate VPS34 to achieve a balanced autophagy activity. Under other cellular stressed conditions or disease state, this balance is disturbed through blocking the UBE3C or TRABID axis, thereby positively or negatively regulating autophagy activity to influence on cell and organism homeostasis.

## Results

**TRABID promotes autophagosome formation.** To explore ubiquitin-dependent regulation of autophagy, we performed an unbiased loss-of-function screen to individually knockdown each of the 92 DUBs for testing their influence on autophagy. To this end, HeLa cells stably expressing Dendra-LC3 were transduced with lentivirus carrying these short-hairpin RNAs (shRNAs), starved, and assayed for the number of Dendra-LC3 puncta using a high-content fluorescent analysis. Among the hits recovered (Supplementary Fig. 1a and Supplementary Data 1), we were particularly interested in TRABID, because of its unique specificity towards the K29- and K33-linked ubiquitin chains[37]. Validation experiment showed that TRABID knockdown reduced autophagosome numbers and LC3 lipidation in cells cultured in Dulbecco's modified Eagle's medium (DMEM; for measuring basal autophagy activity) or Earle's Balanced Salt Solution (EBSS) (for measuring starvation-induced autophagy activity) (Fig. 1a, b). TRABID knockdown also caused the accumulation of autophagic cargo p62 in both conditions (Fig. 1b). Furthermore, in cells treated with bafilomycin A1 to block autophagic turnover, TRABID knockdown still reduced LC3 lipidation but did not alter p62 level (Fig. 1c), indicating a promoting role of TRABID in autophagosome formation. In a reciprocal set of experiments, TRABID overexpression increased autophagosome number in cells cultured in either DMEM or EBSS (Fig. 1d). Accordingly, LC3 lipidation was also enhanced by TRABID overexpression in DMEM- or EBSS-cultured cells treated with bafilomycin A1 (Fig. 1e). These findings collectively identify TRABID as a positive regulator of autophagosome biogenesis.

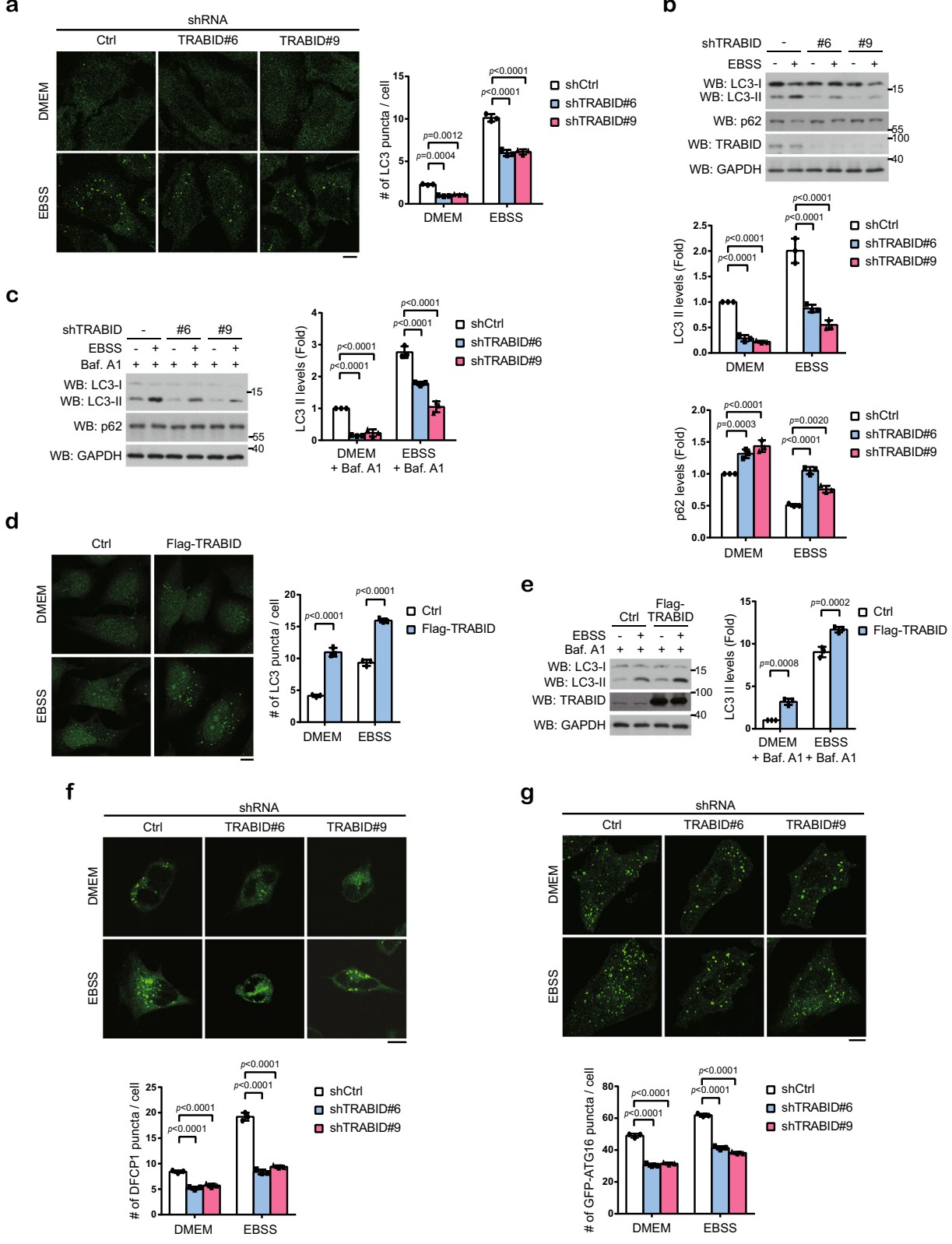

**TRABID reduces VPS34 K29/K48 ubiquitination to stabilize VPS34.** Next, we defined the functional position of TRABID in the autophagosome biogenesis process. TRABID knockdown did not affect mTOR, AMPK and ULK1 activities, as monitored by phosphorylation of ribosomal S6 protein, ULK1 S317 residue, and ATG13 S318 residue, respectively (Supplementary Fig. 1b, c). However, TRABID knockdown diminished DFCP1 puncta,

ATG16 puncta, and ATG14 puncta (Fig. 1f, g and Supplementary Fig. 1d). Based on the positions of these players/markers in the autophagosome formation process[17], our findings suggest an impact of TRABID on the class III PI3-kinase.

Among the subunits of VPS34 complex, TRABID overexpression reduced the ubiquitination level of transfected and endogenous VPS34, but not AMBRA1, Beclin-1, ATG14 and

**Fig. 1 TRABID promotes autophagosome biogenesis. a** HeLa cells stably expressing control or TRABID shRNAs were cultured in full medium or EBSS for 2 h and then analyzed for LC3 puncta by immunofluorescence staining. Representative confocal images are shown on the left and quantitative data are on the right. Bar, 10 μm. The knockdown efficiencies of shRNA are shown in Fig. 1b. **b, c** Western blot analysis of LC3 in HeLa cells stably expressing control or TRABID shRNA cultured in full medium or EBSS for 4 h (**b**) or cultured in EBSS with 100 nM bafilomycin A1 for 2 h (**c**). The blots are representative of $n = 3$ independent experiments. **d** Immunofluorescence staining of LC3 in HeLa cells stably expressing Flag-TRABID and cultured in full medium or EBSS for 2 h. Representative confocal images are shown on the left and quantitative data are on the right. Bar, 10 μm. TRABID expression levels are shown in **e**. **e** Western blot analysis of LC3 in cells as in **d** cultured in EBSS with bafilomycin A1 for 2 h. The blots are representative of $n = 3$ independent experiments. **f, g** HeLa derivatives as in **b** were transiently transfected with GFP-DFCP1 (**f**) or GFP-ATG16 (**g**) and cultured in full medium or EBSS for 2 h. GFP-DFCP1 and GFP-ATG16 puncta were analyzed by confocal microcopy. Representative images are shown on the left and quantitative data are on the right. Bar, 10 μm. Data in **a, b, c, d, f, g** are mean ± SD ($n = 3$ independent experiments, and 30 cells per group per experiment were counted for **a, d, f, g**). P-values are determined by two-way ANOVA with Tukey's post hoc test. Source data are provided as a Source Data file.

UVRAG (Fig. 2a, b and Supplementary Fig. 2a–d). However, TRABID catalytically dead mutant (cs) failed to affect VPS34 ubiquitination (Fig. 2a). We then determined the kind of ubiquitin chain that is removed by TRABID. Using each ubiquitin KR mutant, we unexpectedly found that both K29R and K48R mutants attenuated TRABID-induced VPS34 deubiquitination, whereas K29R/K48R double mutant completed abrogated this deubiquitination (Fig. 2c). Accordingly, the K29/K48-only ubiquitin could confer TRABID-induced VPS34 deubiquitination as the wild type ubiquitin (Fig. 2d). The ability of TRABID, but not its cs mutant, to reduce the K48 ubiquitin chain on VPS34 was further validated using a K48 chain-specific antibody (Fig. 2e). These data provide compelling evidence for the ability of TRABID to remove K29- and K48-linked ubiquitin chains from VPS34. Consistent with the deubiquitination effect of TRABID on VPS34, endogenous TRABID interacted with endogenous VPS34 (Fig. 2f). Furthermore, baculovirally purified recombinant TRABID and VPS34 interacted in vitro (Fig. 2g). Using VPS34 that was ubiquitinated in vitro by an E3 ligase known to generate K29/K48 heterotypic ubiquitin chain (see below), we showed that purified TRABID was capable of reducing the total and K48 ubiquitin linkages from this VPS34 in vitro (Fig. 2h). However, consistent with a previous study[37], TRABID purified from cells cultured in DMEM or EBSS could not hydrolyze the K48-linked diubiquitin (Supplementary Fig. 2e). These findings support the formation of K29/K48 heterotypic chain on VPS34, so that the cleavage of proximal K29 linkages would lead to the removal of distal K48 linkages from the substrate. Altogether, our study identifies VPS34 as a substrate of TRABID. TRABID antagonizes VPS34 K29/K48 heterotypic ubiquitination even though it cannot hydrolyze unanchored K48 diubiquitin chains[37].

We next investigated the functional consequence of TRABID-mediated VPS34 deubiquitnation. TRABID knockdown decreased VPS34 protein abundance and half-life, whereas TRABID overexpression enhanced VPS34 expression and stability (Fig. 2i–l). VPS34 mRNA level, however, was not affected by TRABID knockdown (Supplementary Fig. 2f). Furthermore, the proteasome inhibitor MG132 increased VPS34 expression in control cells but not TRABID overexpressing cells (Fig. 2m). Collectively, our findings suggest a role of TRABID in VPS34 K29/K48 deubiquitination and stabilization.

**TRABID promotes autophagosome maturation.** VPS34 is present in different complexes. The complex I and complex II contain specific subunit ATG14 and UVRAG and are responsible for autophagosome formation and maturation, respectively[38–40]. Immunoprecipitation analysis revealed the association of TRABID with both ATG14 and UVRAG, along with the common subunits of VPS34 complex, in cells cultured in DMEM or EBSS (Supplementary Fig. 2g). Furthermore, GFP-ATG14 and GFP-UVRAG interacted with endogenous TRABID in transfected cells

and these associations were not affected by starvation (Supplementary Fig. 2h). However, no association of TRABID with Rubicon, which forms a VPS34 complex with autophagy-inhibitory function[41], could be detected (Supplementary Fig. 2i). These findings indicate that TRABID is recruited to VPS34 complex I and II. The former is expected by TRABID's function to potentiate autophagosome biogenesis, whereas the latter suggests its role in autophagosome maturation. To test the effect of TRABID on autophagosome maturation, we used the dual-tagged LC3 (RFP-GFP-LC3) to distinguish autophagosome (yellow) from autolysosome (red). We found that TRABID knockdown decreased autophagosome maturation into autolysosome, whereas TRABID overexpression enhanced it (Fig. 2n, o and Supplementary Fig. 2j, k). Thus, besides autophagosome formation, TRABID promotes autophagosome maturation.

**UBE3C assembles K29/K48-branched ubiquitin chains on VPS34.** Having demonstrated the function of TRABID to antagonize VPS34 K29/K48 ubiquitination, we thought to identify the ubiquitin ligase responsible for this ubiquitination event. UBE3C was reported to assemble a K29/K48-branched free ubiquitin chain in vitro[42], but whether this specificity can be recapitulated in its substrates is unclear. Importantly, reciprocal immunoprecipitation demonstrated the interaction of UBE3C with VPS34 endogenously (Fig. 3a). A direct interaction between UBE3C and VPS34 was detected by in vitro pull-down analysis (Fig. 3b). Furthermore, UBE3C depletion diminished VPS34 ubiquitination, whereas overexpression of UBE3C, but not its catalytically dead mutant, enhanced VPS34 ubiquitination (Fig. 3c, d and Supplementary Fig. 3a, b). Liquid chromatography--tandem mass spectrometry (LC–MS/MS) analysis of VPS34 derived from UBE3C-mediated in vivo ubiquitination assay revealed that UBE3C potentiated ubiquitin modification of multiple lysine residues on VPS34 (Supplementary Fig. 3c). The UBE3C-mediated VPS34 ubiquitination was partially inhibited by mutating the ubiquitin K29 or K48 residue, but not other K residues (Fig. 3d). Furthermore, both K29-only and K48-only ubiquitin could partially support UBE3C-induced VPS34 ubiquitination, whereas K63-only ubiquitin could not (Fig. 3e). These findings support a role of UBE3C in mediating VPS34 K29/K48 heterotypic ubiquitination in vivo. To demonstrate that the heterotypic chain contains branched linkages, we compared UBE3C-induced VPS34 ubiquitination level in cells expressing WT, K29/K48R (double mutant), or coexpressing K29R and K48R ubiquitin. While double mutant supports the formation of neither mixed nor branched K29/K48 chains, co-expression of K29R and K48R mutants should enable the formation of mixed but not branched chains. Importantly, UBE3C-mediated VPS34 ubiquitination was greatly compromised by coexpressing K29R with K48R ubiquitin (Fig. 3f). Thus, our data support a role of UBE3C in mediating VPS34 K29/K48-branched ubiquitination.

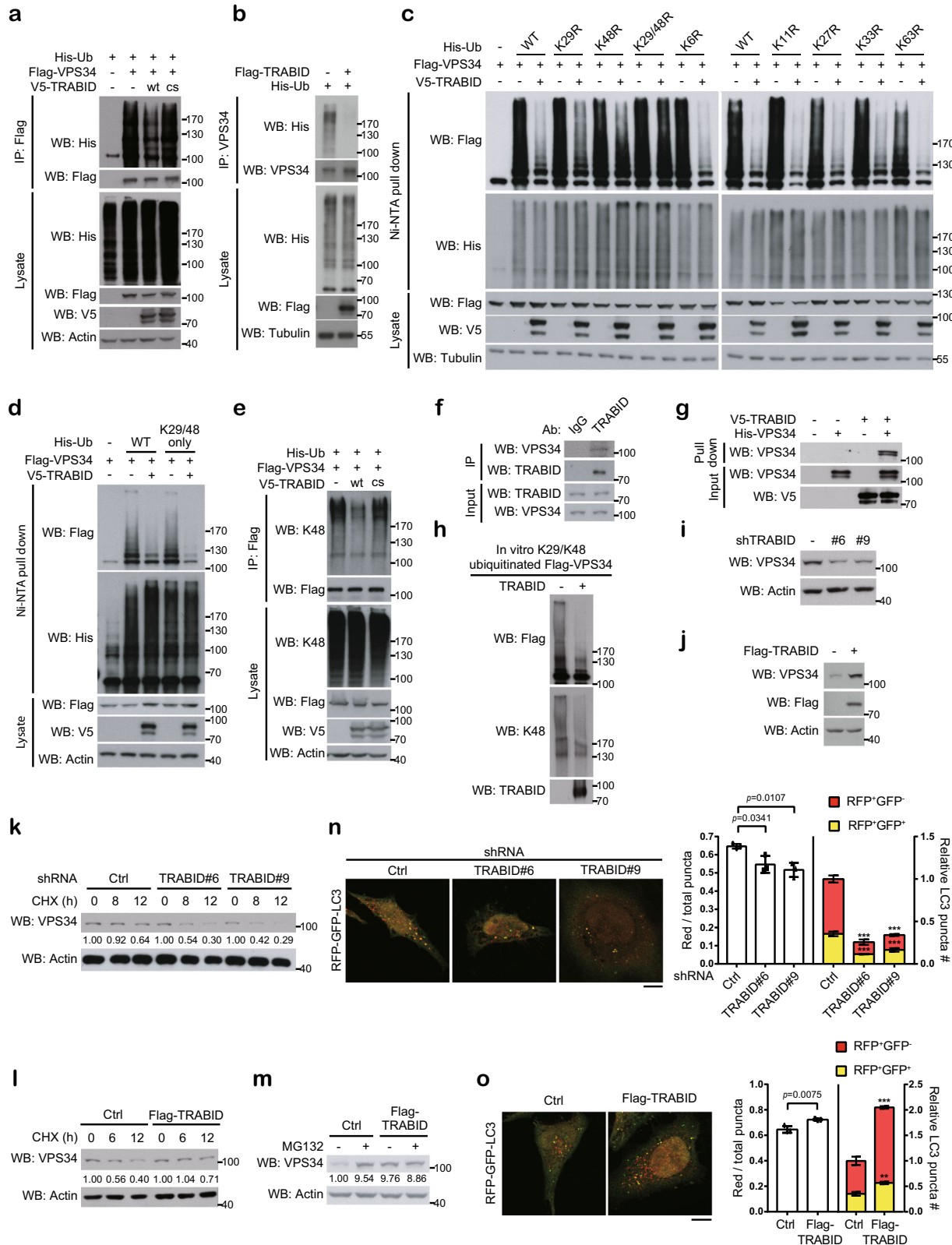

Next, we performed in vitro ubiquitination assay and showed that baculovirally purified UBE3C could assemble polyubiquitin chains on VPS34 purified from 293T cells (Fig. 3g), supporting VPS34 as a direct substrate of UBE3C. To evaluate the formation of branched ubiquitin chains on UBE3C-catalyzed VPS34 and its abundance, we utilized the recently established Ub-clipping method[5]. This method disassembles polyubiquitin chain into GG-modified monoubiquitin by an engineered Lb^pro* protease, which cleaves ubiquitin after the Arg74 residue. Thus, ubiquitin moiety in the linear or branched point generates one or two (or multiple) GG-modified ubiquitin species, respectively, which can be detected by intact mass MS analysis. To validate the methodology, we analyzed the free polyubiquitin chain assembled by UBE3C in vitro. Indeed, after Lb^pro* treatment, intact MS

**Fig. 2 TRABID diminishes the K29/K48 ubiquitination on VPS34 and promotes autophagosome maturation. a, b, e** Western blot analysis of VPS34 ubiquitination (**a, b**) and VPS34 K48 ubiquitination (**e**) in 293T cells transfected with indicated constructs. **c, d** Analysis of VPS34 ubiquitin chain types that are affected by TRABID. The ubiquitinated proteins were pulled down by Ni-NTA agarose under denaturing conditions from 293T cells transfected with indicated constructs and analyzed by Western blot with indicated antibodies. The amounts of different ubiquitin constructs used for transfection were adjusted to allow an equal level of VPS34 ubiquitination among the groups without TRABID transfection. **f** Immunoprecipitation analysis of the interaction between endogenous TRABID and endogenous VPS34 in 293T cells. **g** In vitro interaction of purified TRABID bound on V5-beads with purified His-VPS34. **h** In vitro deubiquitination assay. Flag-VPS34 that was previously ubiquitinated in vitro by UBE3C (as in Fig. 3g) was incubated with or without purified TRABID in the in vitro deubiquitination assay. The reaction mixture was analyzed by western blot. **i, j** Western blot analysis of VPS34 expression in HeLa cells stably expression TRABID shRNA or TRABID construct. **k–m** Western blot analysis of VPS34 in indicated HeLa derivatives treated with cycloheximide for indicated time points or with MG132 for 16 h. **n, o** Immunofluorescence analysis of autophagosome maturation in HeLa-RFP-GFP-LC3 cells stably expressing TRABID shRNAs or TRABID construct cultured in EBSS for 2 h. TRABID expression levels are shown in Supplementary Fig. 2j, k. Bar, 10 μm. The ratios and relative numbers of red puncta to total puncta were quantified and plotted on the right. Data are mean ± SD ($n = 3$ independent experiments, and 10 cells per group per experiment were counted). *P*-values are determined by two-way ANOVA with Tukey's post hoc test (**n** left and **n, o** right; shLuc: yellow vs. shTRABID#6:yellow $P < 0.0001$; shLuc:yellow vs. shTRABID#9:yellow $P < 0.0001$; shLuc:red vs. shTRABID#6:red $P < 0.0001$; shLuc:red vs. shTRABID#9:red $P < 0.0001$; Ctrl:yellow vs. Flag-TRABID:yellow $P = 0.0029$; Ctrl:red vs. Flag-TRABID:red $P < 0.0001$) or unpaired two-sided *t*-test (o left). Source data are provided as a Source Data file.

identified the existence of double GG-modified ubiquitin species and quantification by peak integration indicated that it represents 12.5% of the total ubiquitin (Supplementary Fig. 3d). Next, we analyzed the ubiquitin chain on VPS34 that was assembled by UBE3C in vitro. To this end, VPS34 bound on beads was in vitro ubiquitinated, washed, and cleaved by Lb^pro*. The cleavage product was analyzed by intact MS and LC–MS/MS for detecting branched ubiquitin chains and ubiquitin linkage types, respectively. We found 15.7% of branched ubiquitin chains on UBE3C-modified VPS34 (Fig. 3h and Supplementary Fig. 3e). In addition, K29-GG and K48-GG were the only two types of GG-modified ubiquitin peptides detected and their abundance was greatly enhanced by the presence of UBE3C in the ubiquitination reaction (Fig. 3i and Supplementary Fig. 3f). Together, these in vitro and in vivo studies provide compelling evidence for the ability of UBE3C to assemble K29/K48-branched ubiquitin chains on VPS34.

**VPS34 K29/K48-branched ubiquitination enhances its degradation.** We next determined the consequence of VPS34 branched ubiquitination by UBE3C. UBE3C knockdown in multiple cell lines increased VPS34 protein abundance and inhibited its proteasomal degradation (Fig. 3j, k and Supplementary Fig. 4a). UBE3C knockdown, however, did not affect *VPS34* mRNA level (Supplementary Fig. 4b). UBE3C knockout by the CRISPR-Cas9 strategy also elevated VPS34 expression (Supplementary Fig. 4c). Thus, UBE3C promotes VPS34 degradation. Studies with the K11/K48-branched ubiquitination indicated that such branched chain offers a stronger proteolytic signal than the K48 homotypic chain, which is resulted from an enhanced binding of branched ubiquitin chain to the proteasome[4,9]. To determine whether this is the case for K29/K48-branched ubiquitin chain, we took advantage of the ubiquitin replacement system[43]. In particular, we used the K29R cells, by which doxycycline treatment results in the replacement of endogenous ubiquitin with K29R ubiquitin, thus facilitating a switch of K29/K48-branched ubiquitination to K48 ubiquitination (Fig. 3l). Importantly, we found that doxycycline treatment of K29R cells, but not wild type cells, led to a decreased binding of ubiquitinated VPS34 to the proteasome ubiquitin receptor S5A, also known as RPN10 (Fig. 3m). Accordingly, doxycycline treatment of the K29R cells increased VPS34 stability and attenuated VPS34 proteasomal degradation (Fig. 3n, o). In UBE3C-depleted K29R cells, doxycycline did not alter VPS34 turnover and VPS34 was further stabilized (Fig. 3n). Thus, although VPS34 modified by K48-linked ubiquitin chains is designated for proteasomal degradation, formation of K29/K48-

branched ubiquitin chains further enhances this degradation through an increased binding to proteasome.

**UBE3C and TRABID coordinately govern a balanced autophagy activity.** The ability of UBE3C to induce VPS34 branched ubiquitination and degradation suggests its autophagy-inhibitory effect. Indeed, UBE3C knockdown or knockout increased LC3 puncta and LC3 lipidation in cells cultured in DMEM or EBSS (Fig. 4a, b and Supplementary Fig. 4d, e) and these effects were preserved in bafilomyin A1-treated conditions (Fig. 4c, d and Supplementary Fig. 4f, g). Similar to TRABID, UBE3C was able to associate with VPS34 complex I and complex II (Supplementary Fig. 4h). Furthermore, UBE3C depletion enhanced autophagosome maturation in DMEM- or EBSS-cultured cells (Supplementary Fig. 4i). Thus, UBE3C elicits inhibitory roles in autophagosome formation and maturation.

To demonstrate the antagonizing roles of UBE3C and TRABID in VPS34 and autophagy regulation, we established UBE3C and TRABID double knockdown cells (Fig. 4e). Remarkably, UBE3C depletion-induced upregulation of VPS34 expression, LC3 puncta and LC3 lipidation in cells cultured in DMEM or EBSS were all reversed by TRABID knockdown (Fig. 4e–g). Thus, our data indicate that UBE3C and TRABID coordinately regulate VPS34 stability in basal and starvation conditions, which is crucial for maintaining a balanced autophagy activity.

**ER and proteotoxic stresses attenuate the action of UBE3C on VPS34.** Next, we explored whether the TRABID- or UBE3C-mediated VPS34 regulation could be altered by certain autophagy stimuli. ER stress and proteotoxic stress are known to induce autophagy[44,45]. Compared with basal and starved conditions, the interaction of VPS34 with UBE3C was diminished in response to various ER stressors or proteotoxic stressors, such as tunicamycin, thapsigargin, puromycin, MG132, 17-AAG or 17-DMAG, the latter two being the HSP90 inhibitors (Fig. 5a). The interaction of TRABID with VPS34, however, was not altered under these conditions (Supplementary Fig. 5a). These ER and proteotoxic stressors also decreased the ubiquitination levels of transfected and endogenous VPS34 and increased VPS34 protein abundance (Fig. 5b–d and Supplementary Fig. 5b). Of note, these ER and proteotoxic stressors were capable of inducing autophagy, as evident by increased LC3 lipidation (Supplementary Fig. 5c). Our findings suggest that the balance between UBE3C and TRABID on VPS34 regulation is disturbed by various ER/proteotoxic stressors to favor TRABID-mediated VPS34 stabilization, thus facilitating autophagy induction.

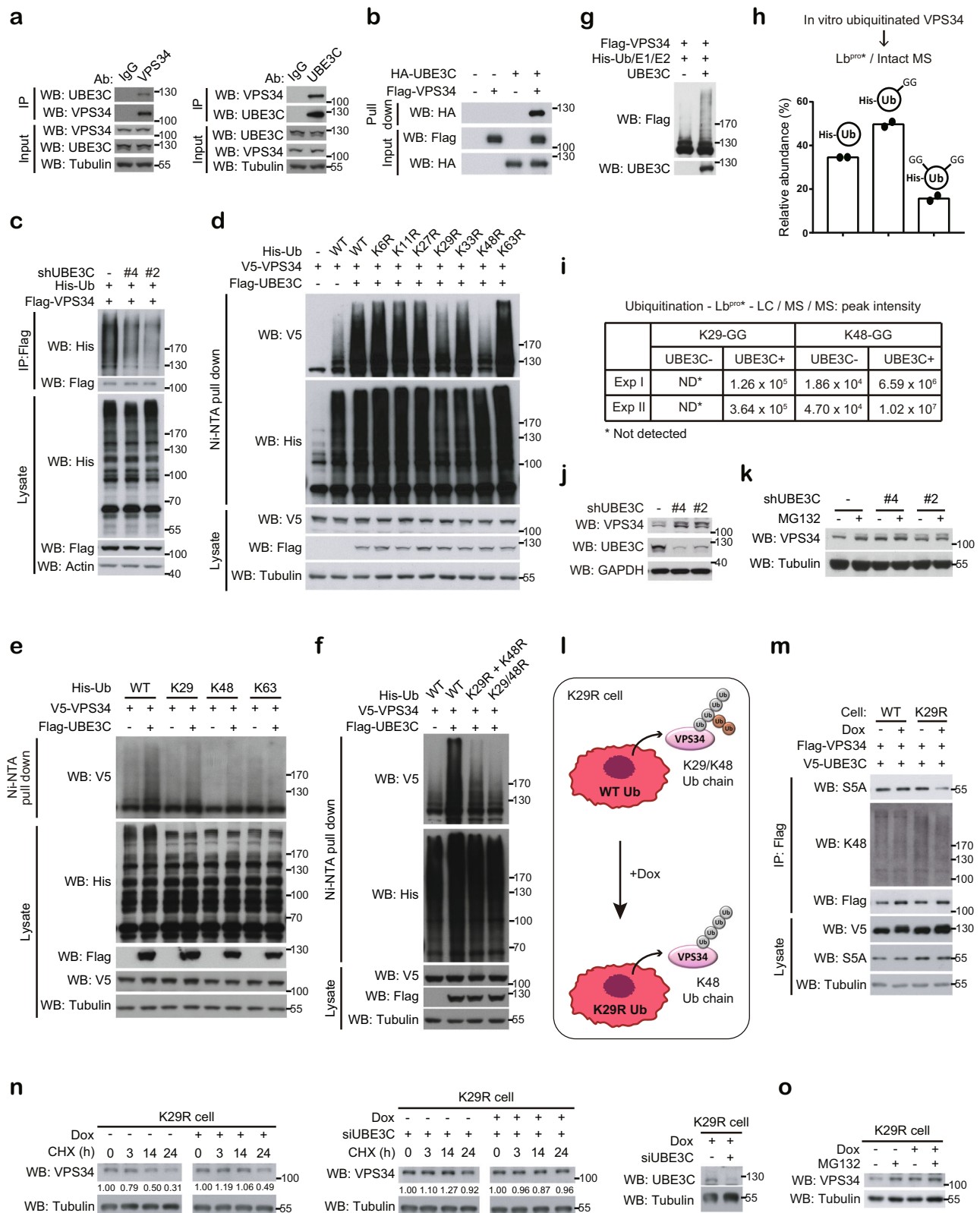

We sought to understand how ER and proteotoxic stresses diminish UBE3C binding to VPS34. In unstressed or starved cells, both TRABID and UBE3C exhibited a remarkable colocalization with ATG16 puncta, which represent phagophores[46] (Fig. 5e, f and Supplementary Fig. 5d upper panel). Given the localization of VPS34 complex I to phagophore[46], this finding is consistent with a concerted action of UBE3C and TRABID on VPS34 regulation

under these circumstances. However, upon tunicamycin treatment, UBE3C showed a poor colocalization with ATG16 puncta, while TRABID recruitment to phagophores was retained (Fig. 5g, h). This finding was recapitulated by using three different fluorescent tags to mark UBE3C, TRABID and ATG16 in the same cell (Supplementary Fig. 5d lower panel). Furthermore, similar findings were observed from puromycin-treated cells, in

**Fig. 3 UBE3C induces VPS34 K29/K48-branched ubiquitination, leading to an enhanced VPS34 proteasomal degradation. a** Reciprocal immunoprecipitation analysis of the interaction between UBE3C and VPS34 in 293T cells. **b** In vitro interaction of baculovirally purified Flag-VPS34 with separately purified HA-UBE3C. **c** Analysis of VPS34 ubiquitination in 293T cells transfected with indicated constructs. VPS34 was immunoprecipitated by M2 beads, followed by western blot analysis with indicated antibodies. The knockdown efficiencies of UBE3C shRNAs are shown in **j**. **d–f** Analysis of VPS34 ubiquitination in 293T cells transfected with indicated constructs. **g** In vitro ubiquitination of Flag-VPS34 purified from 293T cells in the presence or absence of baculovirally purified UBE3C. The reaction mixture was analyzed by Western blot with indicated antibodies. **h** Quantification of indicated ubiquitin species identified by intact MS analysis on in vitro ubiquitinated and $Lb^{pro*}$-treated VPS34. Data are means from two independent experiments. The representative spectrum of intact MS data is shown in Supplementary Fig. 3d. **i** Ubiquitin linkage types and their abundance identified by LC–MS/MS on VPS34 ubiquitinated in vitro in the presence or absence of UBE3C and treated with $Lb^{pro*}$. Peak intensities from two independent experiments are shown. **j, k** Western blot analysis of VPS34 levels in 293T cells stably expressing UBE3C shRNAs (**j**) and treated with MG132 (**k**). **l** Schematic representation for the ubiquitin chain type switch in K29R ubiquitin replacement cells after doxycycline treatment. **m** Immunoprecipitation analysis for the binding of proteasome subunit S5A with ubiquitinated VPS34 in wild type or K29R ubiquitin replacement cells transfected with indicated constructs and treated with or without doxycycline. **n, o** Western blot analysis of VPS34 levels in K29R ubiquitin replacement cells transfected with or without UBE3C siRNA and treated with doxycycline for at least 96 h together with cycloheximide for indicated time points (**n**) or MG132 for 16 h (**o**). The knockdown efficiency of UBE3C siRNA is shown on the right panel of **n**. Source data are provided as a Source Data file.

which TRABID showed a higher colocalization with ATG16 puncta than UBE3C (Fig. 5i, j). Accordingly, while TRABID/ATG16 double-positive puncta were increased in response to tunicamycin or puromycin treatment, UBE3C/ATG16 double-positive dots were decreased (Fig. 5k). In cells treated with proteotoxic stressor 17-AAG or MG132, the impairment of phagophore targeting of UBE3C, but not TRABID, was also observed (Supplementary Fig. 5e, f). Thus, the reduced association of UBE3C with VPS34 under ER and proteotoxic stresses is likely due to a spatial separation of the two proteins.

A portion of UBE3C is associated with proteasome[47] and this association is enhanced under conditions that lead to a general increase of cellular ubiquitinated proteins[48]. In addition, acute induction of a single misfolded protein is sufficient to enhance UBE3C association with proteasome[49]. Since ER and proteotoxic stresses can all increase cellular misfolded and ubiquitinated proteins, we investigated their effects on UBE3C association with proteasome. Indeed, agents that trigger ER or proteotoxic stress all enhanced the association of UBE3C with proteasome, as detected by coimmunoprecipitation of UBE3C with proteasome subunit S5A (Fig. 5l). Altogether, our findings are consistent with a notion that ER and proteotoxic stresses induce a switch of UBE3C localization from phagophore to proteasome, thereby relieving its inhibitory effect on autophagy.

**Enforced targeting of UBE3C to VPS34 attenuates autophagy**. To substantiate that the reduced association of UBE3C with VPS34 contributes to autophagy induction under ER and proteotoxic stresses and to explore the physiological impacts of this regulation, we sought to block this regulation by enforced targeting of UBE3C to VPS34. To this end, we utilized a chemically induced dimerization strategy[50]. We transfected UBE3C knockout cells with constructs for FKBP12-UBE3C and FRB-VPS34 fusion proteins (termed targeting cells). In the control experiment, cells were transfected with FKBP12-UBE3C and VPS34 (without fusion with FRB; termed control cells). As expected, when the targeting and control cells were treated with rapalog together with tunicamycin or puromycin, a stronger interaction between transfected UBE3C and VPS34 was observed in the targeting than control cells and this difference was not seen in cells without rapalog treatment (Supplementary Fig. 6a). Accordingly, rapalog treatment reduced VPS34 steady-state level in puromycin or tunicamycin-treated targeting cells but not in control cells (Fig. 6b). Thus, we established a system that rescues the association between UBE3C and VPS34 upon ER and proteotoxic stresses.

Next, we utilized this system to assess the impacts of ER/proteotoxic stress-induced UBE3C relocation on autophagy

activity and cell homeostasis. Since puromycin induces protein aggregate termed ALIS (aggresome-like induced structure), which is marked by ubiquitin, recognized by ubiquitin-binding autophagy receptors such as p62, and removed by aggrephagy[44,51], we thus evaluated aggrephagy activity in targeting and control cells treated with puromycin. First, rapalog induced a reduction in the colocalization of p62 puncta with LC3 puncta in puromycin-treated targeting cells but not control cells (Fig. 6c), indicating that enforced targeting of UBE3C to VPS34 diminished aggrephagy activity. To evaluate the impact of this decreased aggrephagy activity on the clearance of ALIS, we treated targeting and control cells with puromycin, washed out puromycin, and then induced UBE3C/VPS34 interaction by rapalog during the recovery phase (Fig. 6d, upper right panel). Remarkably, while control cells showed a significant decrease of insoluble ubiquitinated proteins, ubiquitin positive aggregates, and ubiquitin/p62 double-positive dots after 4 h of recovery from puromycin treatment, ALIS clearance was impaired in the targeting cells (Fig. 6d and Supplementary Fig. 6b, c). A similar finding was observed by using PROTEOSTAT dye for detecting cellular protein aggregates (Supplementary Fig. 6d). Thus, enforced targeting of UBE3C to VPS34 impairs the clearance of puromycin-induced protein aggregates via selective autophagy. Besides inducing protein aggregates, ER stress could also stimulate ER-phagy through ubiquitin-dependent and independent mechanisms[52,53]. We therefore determined the impact of ER stress-induced dissociation of UBE3C from VPS34 on ER-phagy using an ER-phagy flux reporter, ssRFP-GFP-KDEL[54]. Once ER fragments are translocated to the lysosome via ER-phagy, the linker between RFP and GFP is cleaved and the GFP fluorescence is quenched to result in red puncta. We showed that enforced targeting of UBE3C to VPS34 in tunicamycin-treated cells led to a reduction of red puncta by confocal analysis and a decrease of cleaved RFP by Western blot analysis (Fig. 6e, f). Importantly, bafilomycin A1 treatment completely blocked reporter cleavage, confirming a lysosome-dependent process. Importantly, upon rapalog treatment, the targeting cells were more susceptible to puromycin- or tunicamycin-induced apoptosis than the control cells, but this difference in apoptosis was not seen in the absence of rapalog (Fig. 6g, h and Supplementary Fig. 6e, f). Altogether, our data support an idea that ER/proteotoxic stress-induced dissociation of UBE3C from VPS34 facilitates the induction of selective autophagy for protein and ER quality control, thereby maintaining cell survival.

**Dysregulation of TRABID/VPS34 axis contributes to liver steatosis**. Next, we sought to explore the pathological impact of VPS34 K29/K48-branched ubiquitination. Autophagy plays an

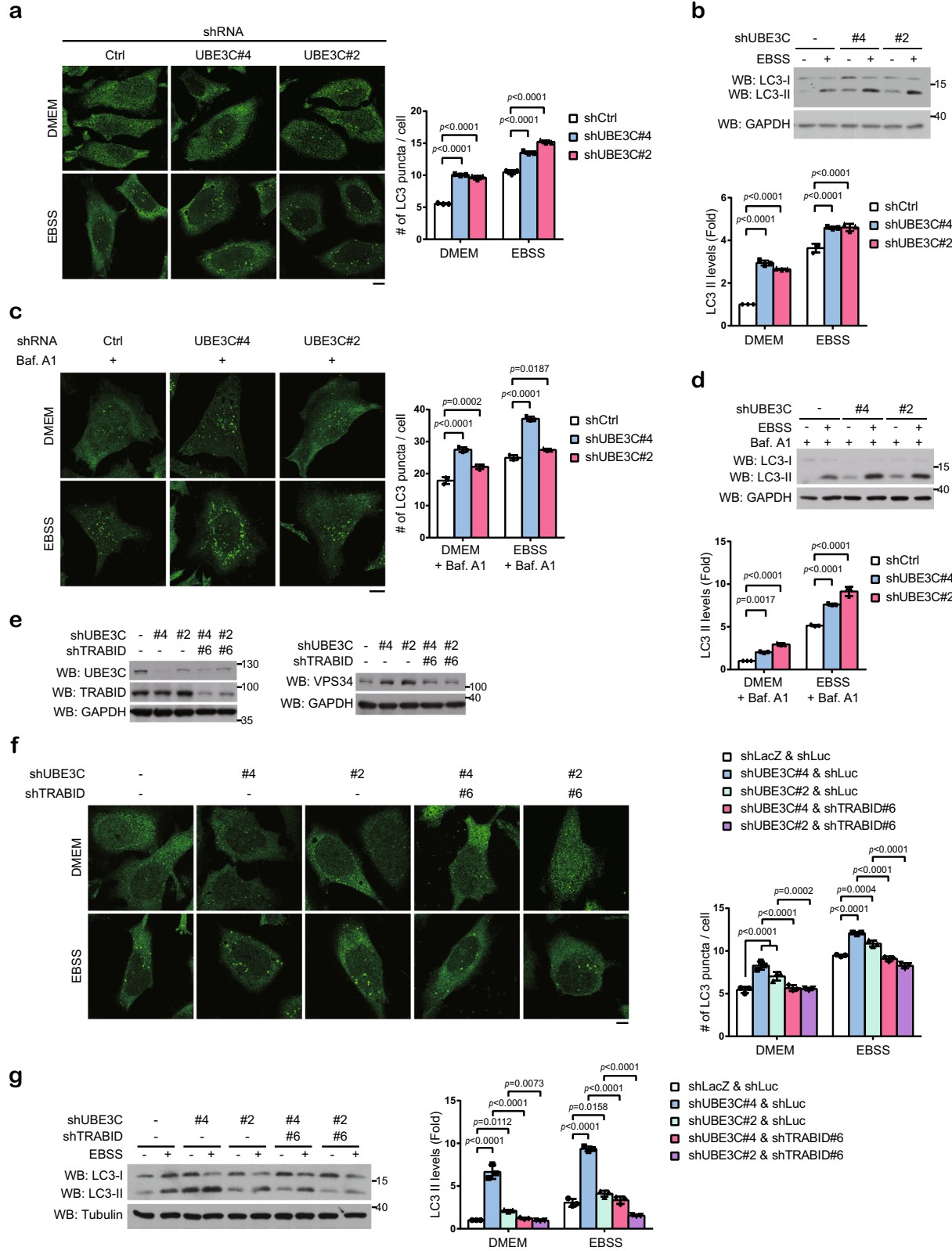

important role in liver metabolism and is responsible for the turnover of lipid droplets to prevent liver steatosis[55]. Recent studies identified multiple links of liver steatosis to autophagy deficiency and ER stress, thereby constituting an intricate vicious cycle to aggravate the disease state[56,57]. We reasoned that elevation of autophagy by VPS34 deubiquitination/stabilization should disrupt the detrimental cycle to alleviate steatosis. To interrogate

the in vivo function of TRABID-mediated VPS34 stabilization in liver metabolism, we set up a mouse model of nonalcoholic fatty liver disease (NAFLD) via feeding mice with a high-fat diet (HFD). After 12 weeks of feeding, the mice were retro-orbitally injected with recombinant adeno-associated virus (rAAV) expressing TRABID or a control vector and sacrificed at 4 weeks later (Fig. 7a). As expected, HFD decreased liver autophagy

**Fig. 4 UBE3C and TRABID coordinately govern a balanced autophagy activity. a, c** Immunofluorescence staining of LC3 in HeLa cells stably expressing UBE3C shRNA and cultured in full medium or EBSS for 2 h in the absence (**a**) or presence (**c**) of bafilomycin A1. Representative confocal images are shown on the left and quantitative data are on the right. Bar, 10 μm. The knockdown efficiencies of shRNA are shown in Supplementary Fig. 4a. **b, d** Western blot analysis of LC3 in HeLa derivatives treated as in **a** or **c**, respectively. The blots are representative of $n = 3$ independent experiments. **e** Western blot analysis of UBE3C, TRABID, and VPS34 in HeLa cells stably expressing UBE3C shRNA and/or TRABID shRNA. **f** Immunofluorescence staining of LC3 in HeLa derivatives as in **e** and cultured in full medium or EBSS for 2 h. Representative confocal images are shown on the left and quantitative data are on the right. Bar, 10 μm. **g** Western blot analysis of LC3 in HeLa derivatives treated as in **f**. The blots are representative of $n = 3$ independent experiments. Data in **a**, **b**, **c**, **d**, **f**, **g** are mean ± SD ($n = 3$ independent experiments, and >30 cells per group per experiment were counted for **a**, **c**, **f**). $P$-values are determined by two-way ANOVA with Tukey's post hoc test. Source data are provided as a Source Data file.

activity, as demonstrated by reduced LC3 lipidation and LC3 puncta (Fig. 7b, c). Notably, these changes were associated with downregulation of both VPS34 and TRABID (Fig. 7b). HFD also decreased the expression of *Trabid* mRNA and TRABID down-regulation was further detected in mouse hepatocyte cell line AML12 treated with oleic acid, but not tunicamycin (Supplementary Fig. 7a, b). These findings suggest a role of free fatty acid accumulation in TRABID downregulation and a contribution of TRABID downregulation to NAFLD. In line with the latter notion, 4-week treatment of rAAV-TRABID not only restored VPS34 expression and autophagy activity, but greatly alleviated HFD-induced increase of liver/body weight (Fig. 7d) and lipid accumulation in the liver, which was evident by gross anatomical view of the livers as well as H&E staining and Oil Red O staining of the liver sections (Fig. 7e and Supplementary Fig. 7c). The HFD-induced increase of hepatic triglyceride content was also reversed by rAAV-TRABID (Fig. 7f). Furthermore, administration of rAAV-TRABID attenuated HFD-induced ER stress, as measured by p-IRE1 and p-PERK levels in the liver and mitigated liver damage, as monitored by alanine aminotransferase (ALT) and aspartate aminotransferase (AST) activities in the serum (Fig. 7g–i). To validate that the effects of rAAV-TRABID on NAFLD alleviation are mediated by Vps34, SAR405, a highly specific inhibitor of the PI3-kinase activity of VPS34[58], was co-administrated with rAAV-TRABID (Fig. 8a). Remarkably, SAR405 (Vps34i) reversed the various NAFLD protective effects of rAAV-TRABID, including the reduction in liver/body weight, hepatic lipid content, and serum levels of AST and ALT (Fig. 8b–e and Supplementary Fig. 7d). Thus, our results highlight the importance of TRABID-mediated VPS34 stabilization in maintaining normal liver metabolism and uncover the contribution of TRABID downregulation to the pathogenesis of liver steatosis through VPS34 destabilization and autophagy deficiency.

## Discussion
Our study identifies a K29/K48-branched ubiquitination on VPS34, the catalytic subunit of class III PI3-kinase and a key player of autophagy. This ubiquitination leads to an enhanced proteasomal degradation of VPS34 and is reciprocally regulated by ubiquitin ligase UBE3C and DUB TRABID. In basal and starved conditions, UBE3C and TRABID coordinately regulate VPS34 ubiquitination to govern a balanced autophagy activity. However, under ER and proteotoxic stressed conditions, this balance is disturbed by a reduced recruitment of UBE3C to phagophore, which is likely a consequence of its enhanced association with proteasome. We provide evidence that this switch of UBE3C localization plays a critical role for cells to cope with the stressed conditions by favoring TRABID-mediated VPS34 stabilization and autophagy induction, as prevention of this switch by an enforced association of UBE3C with VPS34 impairs autophagy and compromises proteostasis, ER quality control and cell survival. In the liver, TRABID-mediated VPS34 stabilization is important for lipid metabolism and is

downregulated in a mouse model of NAFLD. Thus, VPS34 K29/K48-branched ubiquitination can be positively or negatively regulated from the axis of UBE3C or TRABID under different physiological or pathological conditions, thereby inhibiting or stimulating autophagy activity to impact on cell and tissue homeostasis.

Given that the cellular K29 ubiquitin chains mainly exist as the K29/K48-branched chains[13,14] and that the K29 linkage is rather abundant in cells[5,15], we expect that a large number of proteins is modified by this ubiquitin chain type. With VPS34 as an example, we show that this branched ubiquitination enhances substrate binding to proteasome for degradation, presumably resulted from a multivalent binding mode mediated by the branched conjugates. This finding is consistent with the fate of proteins modified by K11/K48-branched ubiquitination[9,59] and the K29/K48-branched ubiquitination-modified model substrate of yeast UFD pathway[16]. Thus, even though K11/K48 and K29/K48-branched chains are topologically distinct, they both confer improved degradation signals.

We reason that such improved degradation signal may be needed for proteins undergoing timely degradation or proteins that are difficult to be degraded. VPS34 is associated with its complex partners and is localized on membrane, both of which might hinder its translocation to proteasome. K29/K48-branched ubiquitination likely enhances VPS34 interaction with VCP/p97 complex to allow an efficient segregation of ubiquitinated VPS34 from membrane and/or its partners. Additionally, an improved degradation signal could be particularly useful for cells to regulate autophagy activity in a timely and dynamic fashion for a rapid adaptation to various stressed conditions. Of note, the K11/K48-branched ubiquitination is induced in response to a number of proteotoxic stresses[59] and is conjugated to certain model substrates of ER-associated degradation (ERAD) and protein quality control[8,10]. The K29/K48-branched ubiquitination is also implicated in the degradation of ERAD substrates in mammalian cells and yeast[8,16]. Under these circumstances, branched ubiquitin chains can offer a more efficient degradation signal than homotypic chain for a rapid disposal of misfolded/unfolded proteins.

In mammals, UBE3C is involved in the formation of K29/K48 ubiquitin chains on certain ERAD substrate[8], indicating its role in protein quality control. In addition, UBE3C and its yeast ortholog Hul5 are reported to increase proteasome processivity by extending the ubiquitin chains on proteasome-bound substrates[60,61]. Intriguingly, the association of UBE3C to proteasome is enhanced in response to a number of proteotoxic stresses and the acute induction of even a single misfolded protein[48,49]. It is believed that this change in UBE3C distribution could enhance proteasomal degradation to alleviate proteotoxic stresses. Besides this beneficial effect, our study reveals an additional advantage of this UBE3C relocation, that is, to relief its inhibitory effect on VPS34 and autophagy, thereby stimulating autophagic degradation of misfolded/unfolded proteins. Through a simultaneous enhancement of proteasome and autophagy activities, both soluble and insoluble misfolded proteins could be targeted for degradation to restore

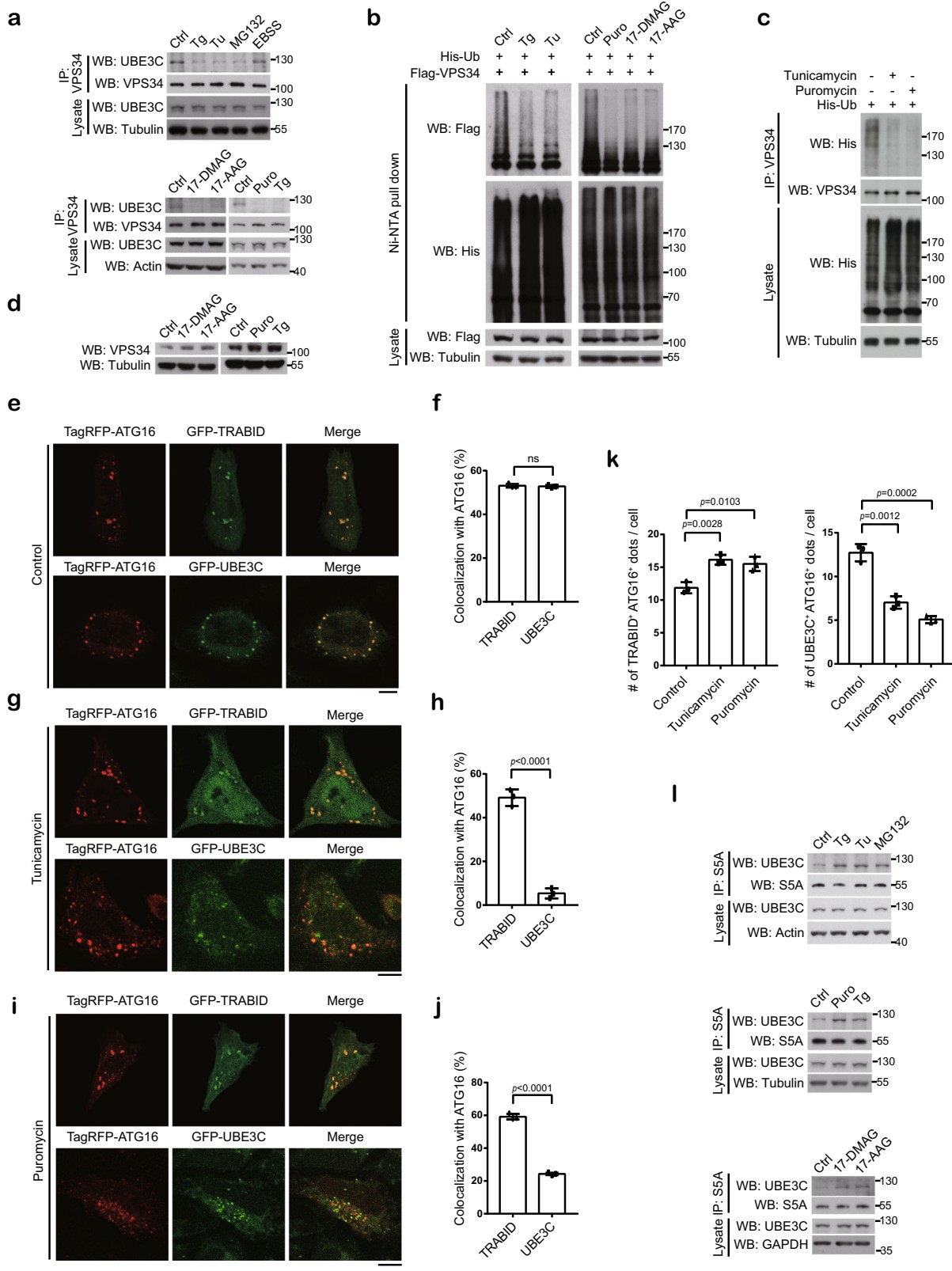

proteostasis. In line with this idea, blockage of this UBE3C relocation by enforced association of UBE3C with VPS34 not only impairs autophagic clearance of protein aggregates but also renders cells more susceptible to proteotoxic stresses. Thus, our study identifies an important function of UBE3C, i.e., sensing the proteasome overload to facilitate autophagy induction. Our findings

further highlight the importance of a concerted function of UPS and autophagy in protein quality control.

Besides its role in modulating proteostasis, VPS34 K29/K48-branched ubiquitination also impacts on liver metabolism. Intriguingly, the expression of TRABID, an enzyme antagonizing this ubiquitination, is decreased in hepatic tissues of NAFLD

**Fig. 5 ER and proteotoxic stresses attenuate the actions of UBE3C on VPS34 by altering UBE3C partners. a** Immunoprecipitation analysis of the interaction between UBE3C and VPS34 in 293T cells treated with 500 μM thapsigargin (Tg), 10 μg/ml tunicamycin (Tu), 10 μM MG132, 10 μM 17-DMAG, 10 μM 17-AAG, 10 μg/ml puromycin (Puro) or starved (EBSS) for 30 min. To ensure a fair comparison, the amounts of immunoprecipitates loaded for Western blot were adjusted to allow an equal level of VPS34 appearing in treated and untreated cells. **b, c** Analysis of VPS34 ubiquitination in 293 T cells transfected with indicated constructs and treated with indicated agents for 1 h. **d** Western blot analysis of VPS34 levels in 293T cells treated with indicated agents for 1 h. **e, g, i** Confocal analysis of the colocalization of TagRFP-ATG16 with GFP-TRABID or GFP-UBE3C in transfected HeLa cells cultured in full medium (**e**) or treated with tunicamycin (**g**) or puromycin (**i**) for 1 h. Bar, 10 μm. The amount of TagRFP-ATG16 plasmid for transfected was adjusted to make a comparable TagRFP-ATG16 puncta number in UBE3C- and TRABID-expressing cells. **f, h, j** Quantitative data for **e, g,** and **i**, respectively. The percentage of GFP-TRABID or GFP-UBE3C puncta colocalizing with TagRFP-ATG16 puncta was analyzed and plotted. **k** Quantitative data for the absolute numbers of GFP-TRABID/TagRFP-ATG16 and GFP-UBE3C/TagRFP-ATG16 double-positive dots in cells treated with indicated agents. Data in **f, h, j,** and **k** are mean ± SD (n = 3 independent experiments, and 5 cells per group per experiment were counted). P-values are determined by unpaired two-sided t-test. **l** Immunoprecipitation analysis of the interaction between UBE3C and proteasome subunit S5A in 293T cells treated with indicated agents for 30 min. Source data are provided as a Source Data file.

mice, which correlates with downregulations of VPS34 expression and autophagy activity. Administration of rAAV-TRABID alleviates the progression of NAFLD, suggesting a potential of role of TRABID overexpression or activation in NAFLD therapy. Notably, TRABID downregulation is also observed in human hepatocellular carcinoma (HCC)[62], a disease that can be progressed from NAFLD. Future study will be aimed to determine the mechanism of TRABID downregulation in NAFLD and the impact of TRABID-stimulated autophagy activity on HCC suppression.

Although TRABID possesses a specific proteolytic activity towards K29- and K33-linked ubiquitin chains, we show that it efficiently removes K48 linkages from VPS34 both in vivo and in vitro. We deduce that this effect may be ascribed to the localization of at least certain K48 linkages at distal position of the branched chain in relation to the K29 linkages. In this way, the cleavage of K29 linkage would remove the distal K48 linkages from the substrate, even though TRABID does not itself hydrolyze the K48 diubiquitin[37]. Nevertheless, we cannot exclude the possibility that another DUB capable of cleaving K48 chain is associated with TRABID in vivo, thereby assisting the deconjugation of K29/K48-branched ubiquitin chains. Regardless of the mechanism that TRABID reduces K48 linkages from the substrate, the partially overlapping chain type specificity of UBE3C and TRABID, as well as their opposite roles in regulating VPS34, suggest that they function as an E3 ligase-DUB pair to govern the K29/K48 ubiquitination status and the stability of a set of cellular proteins. Given that the cellular K29 chain is much more abundant than K33 chain, TRABID NZF1 domain, which binds specifically to K29 and K33 chains, may be exploited to identify cellular proteins that are modified by K29/K48-branched ubiquitination.

TRABID was reported to antagonize K29- and K33-linked ubiquitination on UVRAG[63]. Although such function was not observed in our study, we do not exclude its existence under certain cellular context. In addition, VPS34 was reported to undergo degradable ubiquitination through the E3 ligases Cul1-FBXL20[33] and Cul3-KLHL20[34], but the exact ubiquitin chain types leading to its proteasomal degradation have not been fully characterized. Given the role of UBE3C in modifying proteasome-associated ubiquitinated proteins, it is possible that UBE3C can function as an E4 to add branched ubiquitin chains onto the pre-ubiquitinated VPS34 to enhance its degradation.

In summary, this study discovers a previously unappreciated K29/K48-branched ubiquitination on the critical autophagy regulator VPS34, identifies the enzymes UBE3C and TRABID to reciprocally control this ubiquitination, elucidates the enhanced proteasomal degradation fate of this ubiquitination and reveals

the impacts of this ubiquitination on ER and protein quality control and liver metabolism.

## Methods

**Plasmids**. Plasmids encoding RFP-GFP-LC3, GFP-DFCP1, GFP-ATG14L, GFP-UVRAG, GFP-ATG16L, and TagRFP-ATG16L were kindly provided by Wei Yuan Yang (Academia Sinica, Taipei, Taiwan). Plasmids encoding His-ubiquitin and its KR and K-only mutants, V5-TRABID, Flag-Beclin-1, and Flag-VPS34 were described previously[34,64]. His-K29/K48-only ubiquitin and His-K29R/K48R ubiquitin, V5-TRABID C443S were generated by site-directed mutagenesis. The complementary DNAs (cDNAs) for V5-VPS34 and His-VPS34 were generated by PCR and subcloned to pRK5 and pET32a, respectively. The cDNA for TRABID was subcloned to pRK5-V5, pEGFP-C1 and pLAS5W.Pneo. The cDNA for UBE3C (NP_055486.2; 1-1083 aa) was synthesized by AllBio Science, Inc. (Taichung, Taiwan) and subcloned to pRK5F, pEBFP-C1, pEGFP-C2 and pVL1392. UBE3C C1051S (cs) was generated by site-directed mutagenesis and subcloned to pRK5F. AMBRA1 cDNA was amplified from mRNA derived from 293T cells by reverse transcription PCR (RT-PCR) and subcloned to pRK5F. Plasmids encoding Flag-FKBP12-UBE3C and V5-FRB-VPS34 were generated by inserting the FKBP12 fragment (from pmCherry-FKBP12-C1; Plasmid #67900; Addgene) and FRB fragment (from pEGFP-FRB; Plasmid #25919; Addgene) into pRK5-Flag-UBE3C and pRK5-V5-VPS34, respectively. Plasmid pCW57-CMV-ssRFP-GFP-KDEL (#128257) was purchased from Addgene.

**Antibodies and reagents**. To generate polyclonal antibodies to TRABID, two TRABID fragments, corresponding to the three Npl4-like zinc finger domains (3NZF; residues 1–200) and the ankyrin-repeat domain (Ank; residues 260-340) were cloned to pET32a to generate 6XHis-tagged recombinant proteins. The recombinant proteins were purified by Ni Sepharose (GE Healthcare) under denaturing conditions and used as antigens. Antiserum production and affinity purification were performed by LTK BioLaboratories (Taipei, Taiwan). Other antibodies used in this study are described in the Supplementary Data 2. Oleic acid, Doxycycline, Bafilomycin A1, Cycloheximide, 17-(allylamino)-17-demethoxygeldanamycin (17-AAG) and 17-Desmethoxy-17-N,N-dimethylaminoethylamino-geldanamycin (17-DMAG) were purchased from Sigma-Aldrich, whereas tunicamycin and thapsigargin were obtained from Cayman Chemical. MG132 was purchased from Calbiochem, and rapalog was from Clontech. Puromycin was obtained from Gibco. VPS34 inhibitor (VPS34i; SAR405) was purchased from MedChemExpress.

**Cell culture and transfection**. HeLa, 293T, and 293FT cells were obtained from ATCC. AML12 cells, originally from ATCC, were a gift from Shiou-Hwei Yeh (Institute of Microbiology, College of Medicine, National Taiwan University, Taipei, Taiwan). HeLa-RFP-GFP-LC3 cells were established by transfection of HeLa cells with the RFP-GFP-LC3 construct followed by fluorescence-activated cell sorting of a low-expression population. HeLa and its derivatives were cultured in minimum essential medium (MEM) supplemented with 10% fetal bovine serum (FBS), 1% penicillin/streptomycin (P/S) and 1 mM sodium pyruvate. 293T and 293FT cells were maintained in DMEM containing 10% FBS and 1% P/S. For nutrient starvation, cells were incubated with EBSS (Earle's Balanced Salt Solution; Sigma-Aldrich). U2OS-derived ubiquitin replacement cells[43] were obtained from Zhijian Chen (UT Southwestern Medical Center) and grown in DMEM containing 10% tetracycline/doxycycline-free FBS (Gibco). Simultaneous knockdown of endogenous ubiquitin and expression of exogenous ubiquitin was performed with the addition of 1 μg/ml doxycycline for 48 h. AML12 cells were cultured in DMEM/F12 supplemented with 10% FBS, 1% P/S, 100 μM dexamethasone (D2915; Sigma-Aldrich), 1X Insulin-Transferrin-Selenium-Ethanolamine (51500056; Gibco), 2 mM L-Glutamine, and 1X MEM Non-Essential Amino Acids (11140-050;

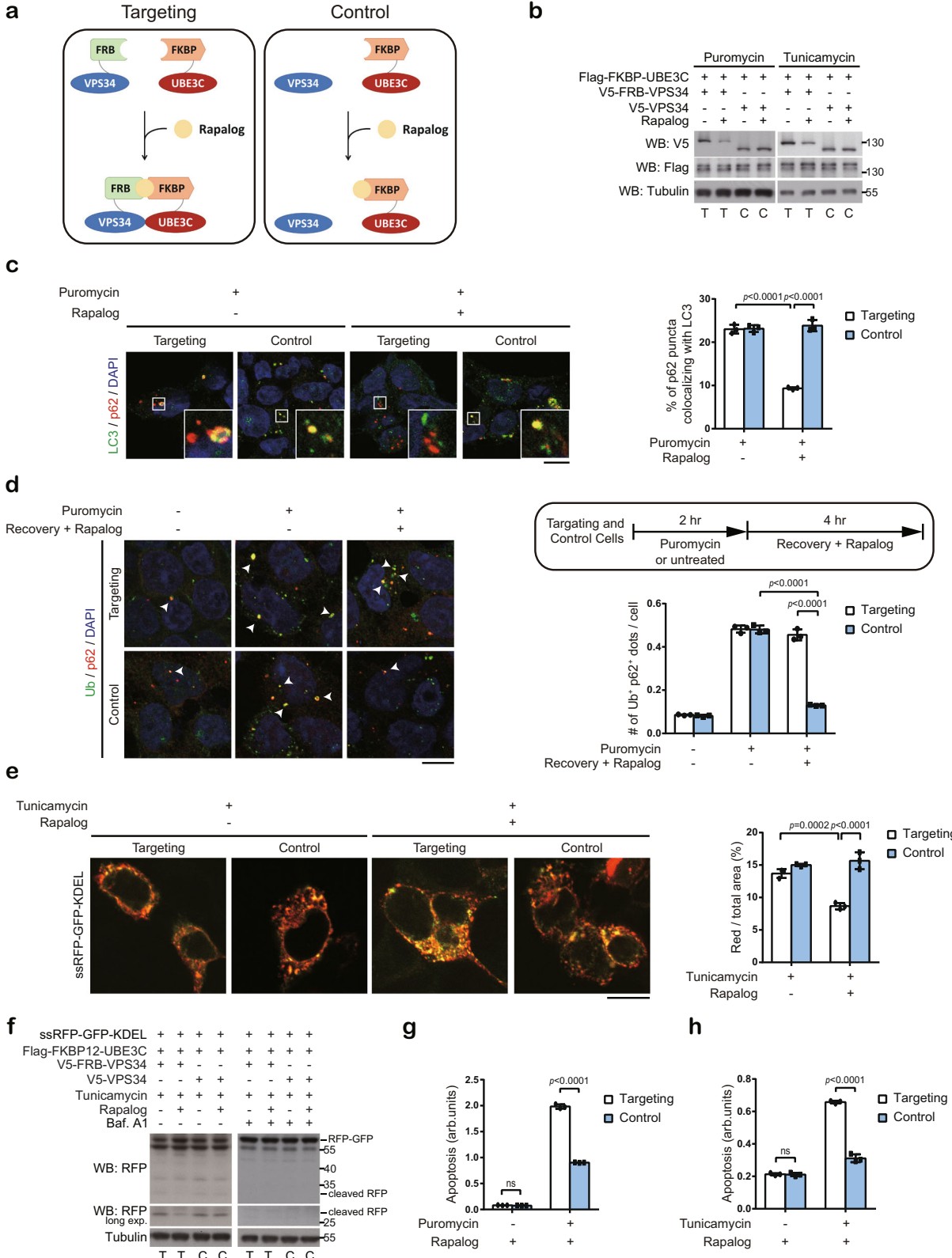

Gibco). Transient transfection of 293FT, 293T and its derivatives was performed using calcium phosphate method, whereas transfection of HeLa cells and HeLa-derived cells were performed with Lipofectamine reagent (Invitrogene).

**Lentivirus production and infection.** 293FT cells were transiently transfected with a packaging mixture containing pCMV Δ8.91 and pMD.G, together with an shRNA construct or a pLAS5w.Pneo-based cDNA construct. The medium was changed at 16 h post transfection and harvested at 20–32 h later. The medium was

filtered through a 0.45 μm syringe filter and supplemented with 8 μg/ml polybrene for infection. The infected cells were selected by 2 μg/ml puromycin or 800 μg/ml neomycin.

**RNA interference.** Lentivirus-based shRNA constructs were obtained from RNA Technology Platforms and Gene Manipulation Core Facility (Taipei, Taiwan). Pooled UBE3C small-interfering RNAs (siRNAs) were purchased from Horizon

**Fig. 6 Enforced binding of UBE3C to VPS34 under ER or proteotoxic stress compromises proteostasis and ER quality control. a** Schematic representation of the experimental design to induce UBE3C and VPS34 association in targeting cells but not in control cells. **b** 293T UBE3C knockout (KO)-derived targeting (T) or control cells (C) were treated with 10 μg/ml puromycin or tunicamycin together with or without 0.5 μM rapalog for 2 h and analyzed by Western blot. **c** Immunofluorescence analysis of 293T UBE3C KO-derived targeting or control cells treated with puromycin and/or rapalog for 2 h. Representative images are on the left and quantitative data are on the right. Boxed regions are enlarged. Bar, 10 μm. Data are mean ± SD ($n = 3$ independent experiments, and 30 cells per group per experiment were counted). $P$-values are determined by two-way ANOVA with Tukey's post hoc test. **d** 293T UBE3C KO-derived targeting or control cells were treated as the scheme outlined and analyzed by immunostaining for ubiquitin/p62 double-positive aggregates. Representative images are on the left and quantitative data are on the right. Arrowheads indicate the double-positive dots. Bar, 10 μm. Data are mean ± SD ($n = 3$ independent experiments, and 30 cells per group per experiment were counted). $P$-values are determined by two-way ANOVA with Tukey's post hoc test. **e** Confocal analysis of 293T UBE3C KO-derived targeting or control cells transfected with ssRFP-GFP-KDEL reporter and treated with tunicamycin and/or rapalog for 6 h. Representative images are on the left and quantitative data are on the right. Bar, 10 μm. Data are mean ± SD ($n = 3$ independent experiments, and 30 cells per group per experiment were counted). $P$-values are determined by two-way ANOVA with Tukey's post hoc test. **f** Western blot analysis of cells as in **e** treated with tunicamycin, rapalog, and/or bafilomycin A1 for 6 h. **g, h** 293T UBE3C KO-derived targeting or control cells as in **b** were treated with puromycin (**g**) or tunicamycin (**h**) together with rapalog for 3 h (**g**) or 6 h (**h**) and analyzed for apoptosis. Data are mean ± SD ($n = 3$ independent experiments). $P$-values are determined by two-way ANOVA with Tukey's post hoc test; ns, not significant. Source data are provided as a Source Data file.

Discovery (Cat# L-007183-00-0005). The target sequences of various shRNAs are listed in Supplementary Table 1.

**Generation of CRISPR knockout cell lines**. UBE3C knockout cells were established by RNA Technology Platforms and Gene Manipulation Core Facility (Taipei, Taiwan). Briefly, two double-stranded oligonucleotides corresponding to the targeting sequences 5′CGGCGGCGCTGCCCGCACAT and 5′CTGGACTCGGGGCCGAGACT located at the exon I of UBE3C gene were cloned to pLAS-CRISPR.Puro, which allows the two sgRNAs are expressed under two independent human U6 promoters. 293T cells were transfected with the resulting plasmid, followed by puromycin selection and single-cell colony isolation. The knockout of UBE3C was confirmed by western blot.

**Reverse transcription quantitative PCR (RT-qPCR)**. Total RNA was extracted using Trizol reagent (Invitrogen). cDNA was synthesized using the iScript cDNA Synthesis Kit (Bio-Rad, Hercules, CA, USA). Quantitative real-time PCR (qPCR) analysis was performed using the SYBR Green PCR Master kit (Applied Biosystems). Amplification was performed on Roche LightCycler 480 system. Gene expression levels were normalized to the housekeeping gene GAPDH. The sequences of qPCR primers are listed in Supplementary Table 2.

**Western blot**. Cells were lysed with RIPA lysis buffer (150 mM NaCl, 20 mM Tris-HCl [pH 7.5], 1% NP40, 0.1% SDS, 1% sodium deoxycholate, 1 μg/ml aprotinin, 10 μg/ml leupeptin, and 1 mM PMSF). Lysates containing equal amount of proteins were resolved by sodium dodecyl sulfate polyacrylamide gel electrophoresis (SDS-PAGE) and proteins were transferred to polyvinylidene fluoride membranes (Millipore). The membranes were incubated with blocking buffer containing 1% BSA or 1–5% non-fat dry milk in TBST (Tris-buffered saline with 0.1% Tween-20) at room temperature for 30–60 min and then incubated with primary antibodies diluted in blocking buffer at 4 °C for overnight. Next, the membranes were washed three times for 10 min each with TBST and then incubated with horseradish peroxidase (HRP)-conjugated secondary antibodies diluted in blocking buffer at room temperature for 1 h. After three times of 10 min-wash with TBST, the HRP signal was detected by Western Lightning™ Plus-ECL (PerkinElmer Inc) or Luminata™ Crescendo (Millipore).

**Immunoprecipitation**. Cells were lysed with RIPA lysis buffer supplemented with phosphatase inhibitors (1 mM Na₃VO₄, 2 mM NaF and 200 μM sodium pyrophosphate). Cell lysates containing equal amount of proteins were incubated with anti-Flag agarose beads (M2; Sigma-Aldrich), anti-V5 agarose beads (Sigma-Aldrich) or GFP-Trap agarose beads (Chromotek) at 4 °C for 2 h. Alternatively, cells were lysed with NP40 lysis buffer (150 mM NaCl, 50 mM Tris-HCl [pH 7.5], 1% NP40, 1% sodium deoxycholate, 1 μg/ml aprotinin, 10 μg/ml leupeptin, and 1 mM PMSF) supplemented with phosphatase inhibitors. Cell lysates were pre-cleared with protein A Sepharose (GE Healthcare) at 4 °C for 1 h and incubated with various antibodies at 4 °C for overnight, followed by 2 h of incubation with protein A Sepharose at 4 °C. After washing the beads with lysis buffer for three times, proteins bound on beads were analyzed by western blot.

**Immunofluorescence**. Cells seeded on coverslips were washed three times with phosphate-buffered saline (PBS) and fixed with 4% formaldehyde at room temperature for 20 min. After three times of wash with PBS, cells were permeabilized with ice-cold methanol for 10 min and then washed three times with PBS. Cells were blocked in PBS containing 1% BSA and 10% goat serum at room temperature for 1 h and incubated with primary antibody diluted in blocking buffer at 4 °C for

overnight. Next, cells were washed three times with PBS, rinsed once with blocking buffer, and then incubated with fluorescent dye-conjugated secondary antibody (Life Technologies) together with DAPI (1 μg/ml) (Sigma-Aldrich) at room temperature for 1 h. Cells were washed three times for 10 min each with PBS and mounted with mounting medium (Dako).

**Confocal microscopy and image analysis**. Cells receiving immunofluorescence staining or cells expressing fluorescent proteins were fixed with 4% formaldehyde at room temperature for 20 min and examined by a confocal microscope (LSM510; Carl Zeiss MicroImaging Inc.) equipped with a 63x/1.40 oil objective lens (Plan-Apo-chromat, Zeiss). The images were collected by a Carl Zeiss ZEN 2009 Laser Scanning Microscope LSM510 (Release Version 5.5.0.0) software. In some cases, cells were examined by a confocal microscope (Olympus FV3000) equipped with a 60x/1.40 oil objective lens (Olympus Objective Lens, PlanApo N) and images were collected by an OLYMPUS FV3000 FV31S-SW (v 2.40) software. To quantify the colocalization of puncta, images were thresholded for particle identification using the Analyze Particles function in Image J 1.52s. The raw intensities of regions containing both puncta signals were calculated through the Measure function of Image J 1.52s. To quantify the area with positive signal, images were thresholded, defined and analyzed by Image J 1.52s.

**shRNA screening and high-throughput image analysis**. HeLa cells stably expressing Dedra2-LC3 were seeded at a density of 1500 cells/well on 96-well plates (Corning® 3603). After cultured for 24 h at 37 °C, cells were incubated with medium containing 8 μg/ml polybrene and lentivirus carrying each of the DUB shRNAs together with a Luc shRNA and an ATG5 shRNA (obtained from RNA Technology Platforms and Gene Manipulation Core Facility) with a multiplicity of infection of 2. The infection medium was replaced with DMEM containing 10% FBS, 1% P/S, and 3 μg/ml puromycin on the next day. At 72 h post-antibiotic selection, stable cell lines were cultured in starvation medium (EBSS) for 3 h. Cells were fixed with 4% paraformaldehyde for 15 min and stained with 1 μg/ml DAPI for 10 min at room temperature. LC3 signal was examined using Cellomics ArrayScan HT fluorescence microscope (Thermo Scientific) with a 20x objective lens. Images were acquired by a Cellomics Spot Detector Bioapplication program and analyzed by Cellomics vHCS:View software version 1.6.2.

**In vitro binding**. For analyzing the in vitro interaction of VPS34 with UBE3C, recombinant HA-UBE3C was purified from baculovirus with anti-HA agarose (Sigma-Aldrich) and eluted with an HA peptide. Flag-VPS34 purified from baculovirus was immobilized on anti-Flag M2 beads and incubated with purified HA-UBE3C in the binding buffer (50 mM Tris [pH 7.5], 150 mM NaCl and 1% NP40) for 30 min. For testing the in vitro binding between TRABID and VPS34, His-VPS34 was purified from bacteria using Ni Sepharose and eluted by imidazole. V5-TRABID purified from 293T cells was immobilized on V5-beads (Sigma-Aldrich) and incubated with purified His-VPS34 in binding buffer for 30 min. In both cases, the beads were washed with binding buffer and the bound proteins were analyzed by western blot.

**In vivo ubiquitination and deubiquitination assays**. Cells transfected with expression construct for His-ubiquitin or its mutant together with other expression constructs were treated with MG132 for 16 h and lysed under denaturing conditions by buffer A (6 M guanidine-hydrochloride, 0.1 M Na₂HPO₄/NaH₂PO₄ [pH 8.0], and 10 mM imidazole). Lysates were incubated with Ni-NTA agarose for 2 h at 4 °C. The beads were washed three times with buffer A/TI [1 vol buffer A: 3 vol buffer TI (25 mM Tris-HCl, pH 6.8 and 20 mM imidazole)] and five times with buffer TI, followed by Western blot analysis. Alternatively, cells transfected with

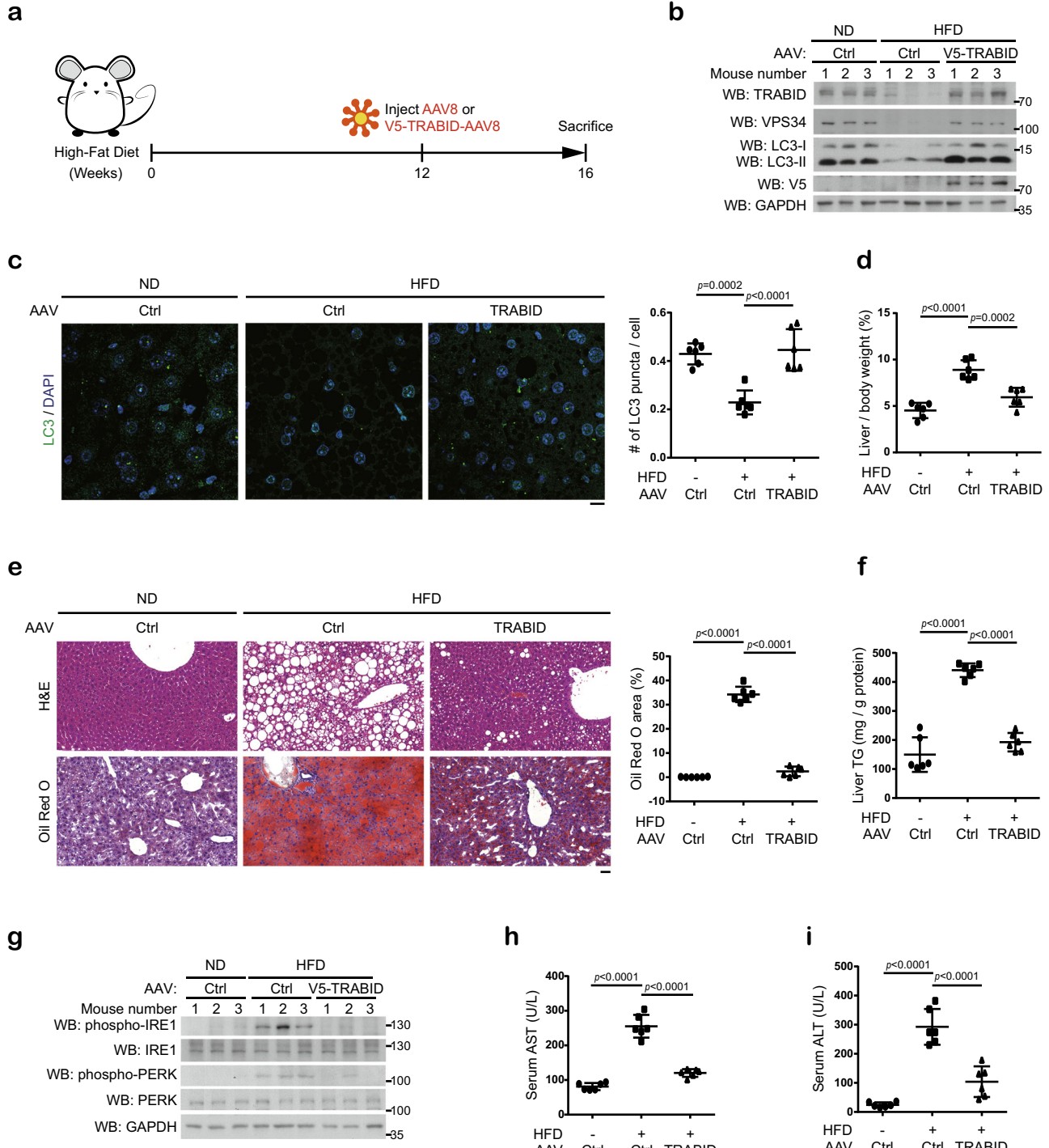

**Fig. 7 TRABID-mediated VPS34 stabilization controls liver metabolism to alleviate NAFLD. a** Schematic representation of the experimental design to evaluate the role of TRABID-mediated VPS34 regulation in NAFLD mouse model. **b**, **g** Western blot analysis of the expression levels of indicated proteins in liver tissues taken from mice on normal diet (ND) or high-fat diet (HFD). **c** Immunofluorescence staining of LC3 in liver sections from indicated mice. Representative images are on the left and quantitative data are on the right. Bar, 10 μm. Data are mean ± SD ($n = 6$, 40–70 cells/section were counted). *P*-values are determined by one-way ANOVA with Tukey's post hoc test. **d**, **f** The percentage of liver per body weight (**d**) and liver triglyceride (TG) levels (**f**) in indicated mice. Data are mean ± SD from six mice. *P*-values are determined by one-way ANOVA with Tukey's post hoc test. **e** H&E and Oil Red O staining of liver sections taken from indicated mice. Bar, 50 μm. The percentages of area stained positive for Oil Red O were quantified and shown on the right. Data are mean ± SD from six mice. *P*-values are determined by one-way ANOVA with Tukey's post hoc test. **h**, **i** AST and ALT assays in indicated mice. Data are mean ± SD from six mice. *P*-values are determined by one-way ANOVA with Tukey's post hoc test. Source data are provided as a Source Data file.

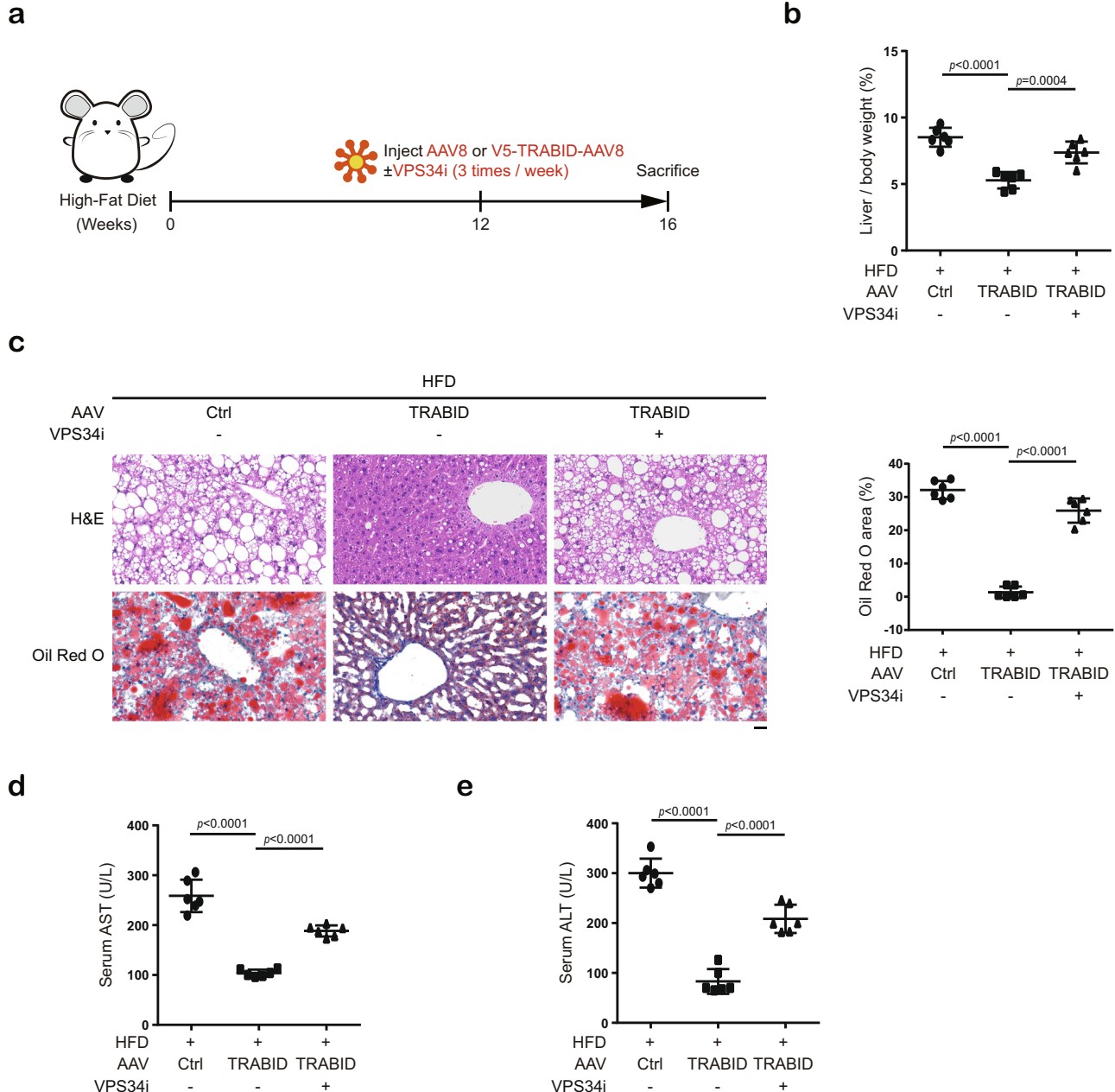

**Fig. 8 VPS34 inhibitor reverses the NAFLD protective effects of TRABID. a** Schematic representation of the experimental design to evaluate whether the NAFLD protective effects of TRABID is mediated by VPS34. **b** The percentage of liver per body weight in indicated mice. Data are mean ± SD from six mice. *P*-values are determined by one-way ANOVA with Tukey's post hoc test. **c** H&E and Oil Red O staining of liver sections taken from indicated mice. Bar, 50 μm. The percentages of area stained positive for Oil Red O were quantified and shown on the right. Data are mean ± SD from six mice. *P*-values are determined by one-way ANOVA with Tukey's post hoc test. **d, e** AST and ALT assays in indicated mice. Data are mean ± SD from six mice. *P*-values are determined by one-way ANOVA with Tukey's post hoc test. Source data are provided as a Source Data file.

Flag-VPS34 together with other constructs were lysed with RIPA lysis buffer. Lysates were subjected to immunoprecipitation with anti-Flag M2 beads, followed by western blot analysis. In all experiments, the equal expression of His-ubiquitin or its variants was checked by western blot.

**In vitro ubiquitination and deubiquitination assays**. In vitro ubiquitination assay was performed in a 20 μl reaction mixture containing 25 mM HEPES (pH 7.5), 200 mM NaCl, 5 mM MgCl₂, 2.5 mM ATP, 50 μM ubiquitin, 200 nM UBE1 (E1), 500 nM UbcH5a (E2), 2 μM UBE3C (E3) (full length, purified from baculovirus) or SUMO-UBE3C$^{HECT}$ (residues 693-1083, purified from *E. coli*), together with or without 500 ng Flag-VPS34 at 37 °C for 40–90 min. For in vitro deubiquitination assay, 10 μl ubiquitinated Flag-VPS34 taken from in vitro ubiquitination reaction mixture was incubated with 2 μM TRABID (Boston Biochem; E-560) at 37 °C for

4 h in 30 μl reaction mixture containing 50 mM Tris (pH 7.6), 50 mM NaCl, and 10 mM DTT.

**Lb$^{pro*}$ purification and treatment**. His-tagged Lb$^{pro}$ (29–184 aa, UniProt ID: P05161) was synthesized and subcloned to pRSF-duet expression vector by Gen-Script (Piscataway, NJ). Lb$^{pro*}$ (L102W) was generated by site-directed mutagenesis to effectively cleave ubiquitin[5]. Lb$^{pro*}$ plasmid was transformed into pLys bacteria strain and the transformants were cultured in LB medium at 37 °C. When the optical density (600 nm) was reached to 0.8, 0.6 mM IPTG was added to induce Lb$^{pro*}$ expression. Lb$^{pro*}$ was purified by nickel affinity chromatography followed by size exclusion chromatography using an SD200 16/60 increase column on an Akta FPLC (GE healthcare, Pittsburgh, PA). Fractions containing Lb$^{pro*}$ were collected, concentrated and flash-frozen by liquid nitrogen. For Lb$^{pro*}$ treatment, 10 mM EDTA was added into the in vitro ubiquitination reaction mixture and

incubated at 37 °C for 10 min. The mixture was treated with 50 μM Lb^pro* in a buffer containing 50 mM Tris (pH 8.0) and 10 mM DTT and incubated at 37 °C for overnight. For intact MS analysis, the mixture was sequentially passed through 50 and 3 kDa Amicon Ultra-0.5 Centrifugal Filters to remove the high molecular weight substances.

**Intact MS for molecular-weight determination**. The protein sample was diluted with 50% acetonitrile and 1% formic acid. An aliquot corresponding to one pmol of the pure protein was injected via the LockSpray Exact Mass Ionization Source (Waters, Milford, MA) with a syringe pump (Harvard Apparatus, MA) and held a flow rate of 3 μl/min throughout the analysis. The mass of intact proteins was determined using Waters Synapt G2 HDMS mass spectrometer (Waters, Milford, MA). The acquired spectra were deconvoluted to single-charge state using MaxEnt1 algorithm of the MassLynx 4.1 software (Waters, Milford, MA).

**LC-MS/MS and label-free quantification**. For determining the polyubiquitin chain types, the Lb^pro*-treated reaction was separated by SDS-PAGE and stained with SYPRO™ Ruby Protein Gel Stain (Invitrogen). The band corresponding to the molecular weight of monoubiquitin was excised for in-gel digestion with Asp-N at 37 °C for overnight. NanoLC−nanoESI-MS/MS analysis was performed on a nanoAcquity system (Waters, Milford, MA) connected to the LTQ Orbitrap Velos hybrid mass spectrometer (Thermo Electron, Bremen, Germany) equipped with a Nanospray Flex interface. Peptide mixtures were loaded onto a 75 μm ID, 25 cm length C18 BEH column (Waters, Milford, MA) packed with 1.7 μm particles with a pore width of 130 Å and were separated using a segmented gradient in 90 min from 5% to 35% solvent B (acetonitrile with 0.1% formic acid) at a flow rate of 300 nl/min and a column temperature of 35 °C. Solvent A was 0.1% formic acid in water. The mass spectrometer was operated in the data-dependent mode. Briefly, survey of full scan MS spectra was acquired in the orbitrap ($m/z$ 350–1600) with the resolution set to 60 K at $m/z$ 400 and automatic gain control (AGC) target at $10^6$. The ten most intense ions were sequentially isolated for HCD MS/MS fragmentation and detection in the orbitrap with previously selected ions dynamically excluded for 60 s. For MS/MS, we used a resolution of 7500, an isolation window of 2 $m/z$ and a target value of 50,000 ions, with maximum accumulation times of 250 ms. Fragmentation was performed with normalized collision energy of 35% and an activation time of 0.1 ms. Ions with singly and unrecognized charge state were excluded.

For determining ubiquitination sites on VPS34, ubiquitinated Flag-VPS34 was isolated by immunoprecipitation with anti-Flag M2 beads from 293T UBE3C knockout cells transfected with Flag-VPS34, His-ubiquitin, together with or without V5-UBE3C under a denaturing condition. The immunoprecipitated proteins were resolved by SDS-PAGE and the protein bands corresponding to the increasing molecular weight of the ubiquitin-conjugated Flag-VPS34 were excised from the gel, and subjected to in-gel digestion with Lys-C and Trypsin at 37 °C for overnight. NanoLC−nanoESI-MS/MS analysis was performed on an EASY-nLC™ 1200 system connected to an Thermo Orbitrap Fusion Luoms mass spectrometer (Thermo Fisher Scientific, Bremen, Germany) equipped with a Nanospray Flex™ ion source (Thermo Fisher Scientific, Bremen, Germany). Peptide mixtures were loaded onto a 75 μm ID, 25 cm length PepMap C18 column (Thermo Fisher Scientific) packed with 2 μm particles with a pore width of 100 Å and were separated using a segmented gradient in 120 min from 5% to 45% solvent B (80 % acetonitrile with 0.1 % formic acid) at a flow rate of 300 nl/min. Solvent A was 0.1% formic acid in water. The mass spectrometer was operated in the data-dependent mode. Briefly, survey scans of peptide precursors from 350 to 1600 $m/z$ were performed at 120 K resolution with a $2 \times 10^5$ ion count target. Tandem MS was performed by isolation window at 1.6 Da with the quadrupole, HCD fragmentation with normalized collision energy of 30, and MS$^2$ scan analysis at 30 K resolution in the orbitrap. The MS$^2$ ion count target was set to $5 \times 10^4$ and the max injection time was 54 ms. Only those precursors with charge state 2–6 were sampled for MS$^2$. The instrument was run in top speed mode with 3 s cycles and the dynamic exclusion duration was set to 15 s with a 10 ppm tolerance around the selected precursor and its isotopes. Monoisotopic precursor selection was turned on.

All data generated were searched against the Swiss-Prot *Human* database, and (561,911 entries total) database using the Mascot search engine (v.2.7.0; Matrix Science, Boston, MA, USA) through Proteome Discoverer (v 2.4.1.15; Thermo Scientific, Waltham, MA, USA). Search criteria used were trypsin digestion, variable modifications set as carbamidomethyl (C), oxidation (M), ubiquitinylation (K) allowing up to two missed cleavages, mass accuracy of 10 ppm for the parent ion and 0.02 Da for the fragment ions. The false discovery rate (FDR) was set to 1% for peptide identifications. Peptide sequence assignments contained in MASCOT search results were validated by manual confirmation from raw MS/MS data. For label-free quantification, precursor ions intensities were extracted using Minora Feature Detector node in Proteome Discoverer with a 2 ppm mass precision and 2 min retention time shift (align the LC/MS peaks mapping to the isotope pattern and retention time).

**Chemical-induced dimerization**. Cells were transiently transfected with Flag-FKBP12-UBE3C together with V5-VPS34 or V5-FRB-VPS34. Dimerization between the FRB- and FKBP12-based fusion proteins was induced by adding 500 nM rapalog (Clontech) to the culture medium.

**Protein aggregate clearance assay**. To evaluate the clearance of ubiquitin aggregates, puromycin-treated cells were washed once with DMEM, cultured in puromycin-free medium for 4 h and examined by confocal microscopy. Alternatively, the clearance of protein aggregates was analyzed using PROTEOSTAT® Aggresome Detection Kit (Enzo Life Sciences). Briefly, cells were fixed, permeabilized and incubated with PROTEOSTAT® dye at room temperature for 30 min. Fluorescence signal was analyzed by a flow cytometer (Beckman CytoFLEX) with a CytExpert version 2.1 software using the 488 nm laser..

**ER-phagy flux assay**. 293T UBE3C knockout cells transfected with ssRFP-GFP-KDEL, Flag-FKBP12-UBE3C together with V5-VPS34 or V5-FRB-VPS34 were cultured in the presence of 0.5 mg/ml doxycycline for 24 h to induce the expression of ER-phagy reporter. Subsequently, cells were washed with PBS for removing doxycycline and then treated with tunicamycin together with or without rapalog for 6 h followed by western blot analysis or confocal microscopy.

**Apoptosis assay**. Cells seeded at a density of $8 \times 10^6$ cells/well in a 6-well plate were treated with puromycin or tunicamycin together with rapalog for 3 or 6 h, respectively. Cells were harvested and DNA fragmentation was measured by Cell Death ELISA Kit (Roche) according to manufacturer's instructions.

**Mouse NAFLD model**. Eight-week-old male C57BL/6J mice purchased from the LASCO CO., Taiwan were fed with a freely available sterilized high-fat diet (DYET #100244, Dyets, Inc.), whose components contain 41.4% of the total calories from fat or with a normal chow diet. For investigating the role of TRABID-dependent autophagy regulation in the liver, mice were retro-orbitally injected with control rAAV8 or rAAV8-TRABID ($1 \times 10^{11}$ vg per mouse, generated by AAV Core Facility in Academia Sinica) and then sacrificed at 4 weeks after injection. For investigating the contribution of VPS34 to the NAFLD protective effect of TRA-BID, 4-week-old male C57BL/6J mice fed with high-fat diet were injected with rAAV8-TRABID and administrated three times a week with VPS34 inhibitor (SAR405) at 10 mg/kg via oral gavage. Mice were sacrificed at 4 weeks later. All animal protocols were approved by Institutional Animal Care and Use Committee, Academia Sinica.

**Histology**. Mouse livers were collected, fixed with 10% formalin buffered with phosphate at 4 °C for overnight, washed and then incubated with 70% ethanol for another overnight. After processing, tissues were embedded in paraffin, sectioned and stained with H&E with a standard protocol.

**Immunofluorescence staining of paraffin-embedded tissues**. Paraffin-embedded tissue sections mounted on slides were deparaffinized at 65 °C for 30 min, incubated in three changes of xylene for 5 min each and rehydrated through graded concentrations of ethanol (100%, 100%, 95%, 85% and 75% for 1 min each). After washed twice with ddH$_2$O for 5 min each, sections were heated in citrate buffer (Scytek) using a BioSB Tinto Retrieve Pressure Cooker and then cooled for 15 min. Next, sections were permeabilized with 0.2% Triton X-100 in PBS for 10 min, washed three times with PBS for 3 min each, blocked with PBS containing 10% goat serum and 1% BSA for 30 min, and incubated with primary antibodies at 4 °C for overnight. After washed three times with PBS for 5 min, slides were incubated with HRP-conjugated secondary antibody (Invitrogen) at room temperature for 30 min. The sections were mounted by mounting medium with DAPI (Santa Cruz).

**Oil Red O staining**. Mouse liver tissues were fixed with ice-cold 4% paraformaldehyde in PBS (Santa Cruz) at 4 °C for overnight. Livers were then incubated with 30% sucrose solution in PBS at 4 °C for overnight. After processed and embedded by optimal cutting temperature (OCT), liver tissues were sectioned, stained with Oil Red O solution and washed with 50% isopropanol and deionized water. Liver sections were counterstained with hematoxylin. The sections were photographed by Pannoramic 250 FLASH II Slide Scanner and analyzed by a 3DHISTECH's CaseViewer software (version 2.3). Quantification of Oil Red O positive area was performed with the Histoquant module in a 3DHISTECH QuantCenter software (version 2.0.0.46136). One area was selected from the slide for each mouse liver and the percentage of area showing positive signal was calculated.

**Serum biochemical analysis**. Blood samples were obtained from facial vein or cardiac puncture before sacrifice. Following serum collection by centrifugation, ALT and AST levels were measured by DRI-CHEM 3500 s (FUJIFILM).

**Hepatic triglyceride level**. Triglyceride in the liver was extracted by homogenizing the tissue with a 1 ml solution of 1:2 methanol/chloroform (v/v), followed by sonication at 37 °C for 30 min and shaking at 4 °C for overnight. After centrifugation to pellet down the debris, 0.25 ml of chloroform and 0.25 ml of water were added to the liquid material and vortexed for 30 min. The lower organic phase was transferred to a new tube and the solvent was

evaporated using a speed vacuum apparatus. The pellet was resuspended with TR0100 reagent (Serum Triglyceride Determination Kit, Sigma), and then incubated for 10 min at room temperature, followed by absorbance measurement at 540 nm.

**Statistics and reproducibility**. The unpaired two-sided Student's $t$-test was used to compare between two groups and the one-way or two-way ANOVA with Tukey's post hoc test was used for multi-group comparison. All statistical analyses were conducted at a significance level of $P < 0.05$ and calculated by using GrapPad Prism v.7. For Western blots or confocal images, data are representative of three (for Fig. 2a–g, i, J; Fig. 3a–d, g, j, k, m; Fig. 4e; Fig. 5a–d, l; Fig. 6b; Supplementary Fig. 1b, 1c; Supplementary Fig. 2a–d, g–k; Supplementary Fig. 3a, b; Supplementary Fig. 4a, c, e, g, h; Supplementary Fig. 5a–f; Supplementary Fig. 7b) or two (for Fig. 2h, k, l, m; Fig. 3e, f; Fig. 6f; Fig. 7b, g; Supplementary Fig. 2e, Supplementary Fig. 3d, e; Supplementary Fig. 6a, b) independent experiments.

**Reporting summary**. Further information on research design is available in the Nature Research Reporting Summary linked to this article.

## Data availability

All data supporting this study are available within this article and Supplementary files. The original Mass Spectrometry data for identifying VPS34 ubiquitination chain types and ubiquitination sites are deposited to the ProteomeXchange Consortium via PRIDE partner repository with the project accession number PXD023986 and PXD023959, respectively. All other data supporting the findings of this study are available from the corresponding author upon reasonable request. Source data are provided with this paper.

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

## Acknowledgements

We thank Wei Yuan Yang for constructs, Zhijian Chen for ubiquitin replacement cells, the Academia Sinica Common Mass Spectrometry Facilities located at the Institute of Biological Chemistry for mass spectrometry analyses, RNA Technology Platforms and Gene Manipulation Core Facility, Taiwan for shRNA constructs, Cellomics ArrayScan analysis, and construction of CRISPR knockout cell lines, Chin-Chun Hung for assisting confocal analysis, AAV Core Facility in Academia Sinica for rAAV generation and Pathology Core Facility at the Institute of Biomedical Sciences for tissue processing and pathology analyses. This work is supported by MOST Academic Summit Grant (108-2639-B-001-001-ASP) and an intramural fund from Institute of Biological Chemistry, Academia Sinica.

## Author contributions

Y.-H.C. conceived the study, designed and performed most experiments, and analyzed the data. T.-Y.H., H.-J.H., and R.-L.Y. performed some confocal and cell-based analyses. Y.-T. L. helped for branched ubiquitination analysis. W.-H.L. helped some animal experiments. H.-W.T. performed shRNA screen. S.-Y.L. conducted mass spectrometry analyses. Z.-Q.S. instructed NAFLD-related experiments. G.-C.C. supervised shRNA screen and provided conceptual advices on autophagy. K.-P.W. provided Lb^Pro* and supervised the branched ubiquitination analyses. T.-F.T. supervised NAFLD studies. R.-H.C. directed and coordinated study, designed the research and oversaw the project. R.-H.C. and Y.-H.C. wrote the manuscript.

## Competing interests

The authors declare no competing interests.
