## [Peer Review File · Nature Communications]

REVIEWER COMMENTS

Reviewer #1 (Remarks to the Author):

In this manuscript, authors investigated the role of E3 ligase UBE3C and deubiquitinase TRABID in the regulation of autophagy, proteostasis and liver lipid metabolism. They provided evidence to show that TRABID deubiquitinates VPS34 K29/K48 to prevent its proteasomal degradation and increased stabilization of VPS34 to promote autophagosome formation. In contrast, the E3 ligase UBE3C promotes the branched ubiquitination and proteasomal degradation of VPS34 to inhibit autophagy. They also showed physiological relevance of this pathway in the context of diet-induced non-alcoholic fatty liver disease (NAFLD). The results overall identified a novel ubiquitination regulation of VPS34 stability in response to autophagy stimuli through the reciprocal regulation by TRABID and UBE3C, which may be critical in maintaining a balanced autophagy activity. Most data are of good quality and manuscript was well written. However, some issues have to be addressed (details listed below) to improve the quality of this manuscript.

Specific comments:

1. The citation on the cross-talk of UPS and autophagy was mainly recent review papers (ref 20-23). Two pioneer original research papers on the cross-talk of ER stress, UPS and autophagy should be cited (PMID: 16901900 and PMID: 17620365).
2. Figure 1c & e, to increased data rigor and reproducibility, the experiments should be repeated for at least 3 times and densitometry analysis data should reflect these repeated experiments as these are the key fundamental data for the main conclusion of this manuscript. Similar requirements should also be applied for Figure 4.
3. Data presentation for RFP-GFP-LC3 puncta assay in Figure 2 m & n was questionable. VPS34 affects the upstream autophagosomes formation/biogenesis (also supported by Figure 1 in this manuscript although it may also regulate autophagosome maturation), then the ratio of red vs total of LC3 puncta (from RFP-GFP-LC3 assay) would be misleading as the possible altered green only puncta numbers by manipulating TRABID. Authors should present the red and yellow puncta numbers separately in addition to the ratio data.
4. Figure 3m-o, the labeling shUb-Ub should be clearly defined in the text or figure legend.
5. Figure 5, various ER stress inducers decreased the interaction of UBE3C with VPS34 and did not change its interaction with TRABID (supplemental Fig 5). While the authors tended to conclude that ER stress favors the stabilization of VPS34 to promote autophagy, data in Figure 5a did not show significant increase of VPS34. This would conflict the conclusion. As most experiments were performed under the overexpression conditions, it is important to show the endogenous ubiquitination and stabilization of VPS34 under the ER stress conditions.
6. Figure 6c, p62 interacts with LC3 via its LIR, the rapalog experiments should only affect the number of GFP-LC3 puncta numbers, it was unclear how it would affect the interaction of p62 with LC3 (green only LC3 puncta that was not colocalized with p62)? In addition to Figure 6d, western blot analysis for protein aggregates (ubiquitination) would be more quantitative and robust.
7. Figure 7, how did HFD decrease TRABID? Did the ER stress inducers in cultured cells also lead to the decreased TRABID? Was the protection from TRABID overexpression against diet-induced NAFLD solely due to increased autophagy? Increased de novo lipogenesis is also implicated in HFD-induced NAFLD. Did TRABID also affect de novo lipogenesis?

Reviewer #2 (Remarks to the Author):

The manuscript by Chen et al. identifies TRABID and UBE3C positive and negative regulators, respectively, of autophagosome formation and maturation. This is a very comprehensive study from Chen et al. The experiments are well carried out and the conclusions are largely supported by the data. The paper would be strengthened by more experiments being done at the endogenous level but

it is understood this can be very difficult to do properly and should not prevent publication.

Comments:

The authors must tabulate the raw data from the shRNA screen presented in supplementary Figure 1a thereby revealing other proteins that potentially regulate LC3 puncta formation. This would be a valuable resource for the community.

Authors should include a sentence explaining why experiments were carried out on both fed and starved cells.

I do not understand the WB in figure 1bc and e. What does the I and II annotation for the two bands mean? Presumably the knockdown efficiencies printed at the bottom of the top panels correspond to TRABID knockdown so why does the key state LC3 II? Author's should explain how the efficiencies were calculated in the figure legend.

Supporting publications should be cited/cited again where appropriate in the results sections. E.g. "These findings suggest an impact of TRABID on the class III PI3-kinase REFs".

Assertions are invariably derived from experiments involving a double overexpression experiment. For example, in Figure 2a it would be much more convincing if endogenous VPS34 was being visualized rather than overexpressing both it and TRABID.

Figure 2d. It is surprising that TRABID has activity towards K48 chains. However, previous specificity studies have been carried out with protein truncated to the OTU and ankrin repeat domains. The authors might want to provide an explanation for why K48 activity might be present in the context of full-length protein. E.g. Maybe the NZF domains bind K29 linkages (as shown by the Komander lab, Michel et al. 2015) and position the catalytic domain in proximity of K48 linkages.

A nice complementary experiment would involve preparing heterotypic K29/K48 chains in vitro using recombinant UBE3C. These could then be tested as a substrate for full-length recombinant TRABID and analyzed using the K48 antibody.

The implementation of the ubiquitin replacement strategy is nice.

Minor comments:

Some of the claims should be toned down. For example line 177 should be changed to- Collectively our findings suggest.....

The following sentence like an oxymoron.

"Together, our study identifies VPS34 as a substrate of TRABID. TRABID antagonizes VPS34 K29/K48 heterotypic ubiquitination even without directly hydrolyzing the K48 ubiquitin chain."

How about:

"Together, our study identifies VPS34 as a substrate of TRABID. TRABID antagonizes VPS34 K29/K48 heterotypic ubiquitination even though it does not hydrolyze unanchored K48 diubiquitin chains (ref.)."

Line 219. Please stick to the terminology defined in the introduction and say "homotypic" rather than "linear".

Reviewer #3 (Remarks to the Author):

In the manuscript "VPS34 K29/K48 branched ubiquitination governed by UBE3C and TRABID regulates autophagy, proteostasis and liver metabolism" by Chen et al, the authors report that ubiquitin ligase UBE3C and deubiquitinating enzyme TRABID reciprocally target VPS34 for K29/K48 branched ubiquitination to control the protein stability of VPS34. There are several concerns (some listed below) that I have with this study. Most of these concerns are associated with slight differences of VPS34 protein levels of most western blot images and some results are not convincing. This study is forced to put the data together and the mouse model of nonalcoholic fatty liver disease (NAFLD) is not sufficient to support the conclusion that VPS34 ubiquitination governed by UBE3C and TRABID regulates liver metabolism. Specific comments are listed below.

Major comments:

1. The quality of image in Figure 1a is quite low as I cannot clearly observe the morphology of the HeLa cells. In Figure 1b-c, the authors had better to detect the expression of TRABID in the same experiment setting, thus providing the evidence that there is a positive correlation between TRABID and autophagosome formation. To further indicate that TRABID promotes autophagy process, the authors should analysis the degradation of autophagic substrate p62. In Figure 1g, we can hardly see any significant difference of the ATG16 puncta in TRABID depletion cell when compared with control samples. I strongly suggest the authors label the expression of TRABID with fluorescent antibody in the confocal microscopy experiments.
2. The authors claimed that TRABID can cleave the K29/K48 branched ubiquitin chains on VPS34 and the ubiquitination enhances the binding of VPS34 to proteasome for degradation, however, in all the co-immunoprecipitation experiments of Figure 2, the protein levels of VPS34 are not up-regulated with TRABID overexpression in the lysate samples. The role of the K29/K48 branched ubiquitination in VPS34 stabilization should be further confirmed. It has been reported that unanchored ubiquitin chains act as critical mediators in the cellular signal transduction, the authors need check whether VPS34 is covalently attached to anchored ubiquitin chains or noncovalently with unanchored ubiquitin chains. Moreover, which lysine site of VPS34 is associated with the K29/K48 branched ubiquitination should be dissected.
3. In the lysate images of Figure 3b, 3c, 3d, 3e and 3f, it seems that overexpression of UBE3C cannot mediate the degradation of VPS34 and knockdown UBE3C has no effect on the protein stability of VPS34. The authors should also resolve the concern that whether UBE3C degrades VPS34 in a ubiquitination modification dependent manner.
4. Does UBE3C or TRABID affect the mRNA levels of VPS34? As VPS34 occurs in two main multiprotein complexes, referred to as complex I and complex II, which are made up of VPS34, VPS15, Beclin-1 and ATG14 (complex I) or VPS34, VPS15, Beclin-1 and UVRAG (complex II). Complex I is activated and recruited to the phagophore initiation sites mainly at the endoplasmic reticulum, to produce a pool

of PtdIns3P critical for the formation and elongation of the nascent autophagosome, which encloses and isolates cytoplasmic components, whereas complex II controls endosome maturation and promotes autophagosome–late endosome/lysosome fusion. Which pool of VPS34 does UBE3C or TRABID alter?

5. In Figure 5c, the authors showed that 17-DMAG, 17-AGG, Puro and Tg increased VPS34 stability, but in the right panel of Figure 5a, the protein abundances of VPS34 in lysate do not change with the pharmaceutical treatments.

6. As TRABID is a positive regulator while UBE3C is a negative regulator of autophagosome formation, why TRABID overexpression does not caused higher amounts of ATG16 puncta when compared with UBE3C-overexpressing group in Figure 5d, 5f and 5h?

7. It has been reported that there is a prominent diminish autophagy flux in NAFLD and several major mechanisms that may account for the impairment of autophagy in NAFLD. Downregulation of autophagy proteins (such as Beclin-1, ATG7, ATG5) (Yang et al. 2010, Cell Metabolism), decreased lysosomal acidification (Wang et al. 2018, FASEB J), defective autophagosome-lysosome fusion (Koga et al. 2010, FASEB J; Park et al. 2014, Nature Communications) may lead to decreased autophagy flux in NAFLD. Does TRABID-controlled liver metabolism to alleviate NAFLD go through VPS34? Does the decreased VPS34 level with HFD treatment associate with Beclin-1 or other ATG proteins? Does VPS34 depletion cause NAFLD? It is important to check the role of TRABID and UBE3C in NAFLD with the deficiency of VPS34.

Minor comments:

The authors had better label the cells with normal cultured condition (DMEM) and starvation (EBSS), but not with Fed, as the experiments are carried out in cells but not animals.

2. In Figure 7j, the authors need remodified their working model, as they have not evidence to demonstrate that UBE3C plays a physiologic role in mice with HFD treatment.

Reviewer #4 (Remarks to the Author):

VPS34 is a critical regulatory component of Beclin1 complex which controls basal and stress-dependent autophagic proteolysis. Although ubiquitin-dependent VPS34 turnover has been well-been characterized, studies have mainly focused on linkage types of ubiquitin chain and simple degradation biology with respect to homotypic ubiquitin chains. This paper written by Chen et al., elucidates mechanisms on regulating autophagic proteolysis via assembly of heterotypic ubiquitin chain on VPS34 turnover by UPS. The authors propose that TRABID and UBE3C are DUB and E3 ligase which remodel ubiquitin chains of VPS34 to generate K29/K48 branched ubiquitin chain. By addition of K29-linked ubiquitin on the basement of the K48 linked ubiquitin chain, VPS34 is targeted to the proteasome compartment with increased affinity. Overall, the major claims are supported by a set of carefully designed experiments. The results from this study provide critical information in the field of heterotypic ubiquitin chains and their mechanistic and functional analysis. Nonetheless, there are some parts that should be addressed for consideration for publication. The revised manuscript will be met with greater enthusiasm if these comments are sufficiently addressed, hopefully with some additional experiments.

Major comments

1. Overall, it remains unclear how the basal turnover of VPS34 is regulated. The results clearly show that VPS34 with K29/K48-linked heterotypic ubiquitin chain is degraded via the proteasome. The authors should address or at least discuss how VPS34 assembled with pre-existing K48-linked homotypic ubiquitin chain, a well-known UPS signal, is degraded. Some previous studies proposed that specific linkage types of ubiquitin chain do not affect VPS34 degradation. Any explanation for this discrepancy?

2. The current study demonstrated that VPS34 regulatory mechanism is redirected to UPS upon

assembly of heterotypic ubiquitin chain. The authors claim that this mechanism represents “a crosstalk of UPS-autophagy via VPS34”. In my opinion, “crosstalk” should involve the communication between UPS and autophagy to achieve proteostasis in the cell. I am not sure whether “crosstalk” can be used in this case.

3. One remaining question arising from this study is the exact lysine residue of VPS34 responsible for heterotypic ubiquitin chain. Is it K419? There are some literatures which propose different E3 ligases responsible for ligating K48-linked ubiquitin chain on VPS34. It is unclear whether UBE3C responsible for assembly of both K48- and K29-linked ubiquitin chain at the identical residue or UBE3C can build K29-linked ubiquitin chain on any of the K48-linked ubiquitin chain basement.

4. The authors proposed that both ubiquitin-dependent and -independent ER-phagy under ER stress are regulated by UBE3C and VPS34 (Fig. 6e). However, a previous study identified the ER transmembrane receptor protein TRIM13 as a K63-linked ubiquitination substrate which recruits autophagic adaptor p62 for ubiquitin mediated ER-phagy. Given that the ER is a site of omegasome generation, co-localization assay of LC3 with the membrane protein calnexin is not sufficient to claim that the UBE3C-VPS34 pathway mediates ubiquitin-dependent selective ER-phagy. The author may need to clarify this point by assessing the role of UBE3C-VPS34 pathway on the autophagic co-localization and turnover of ER-resident luminal proteins. Also, autophagic flux assay should be correctly performed by comparing bafilomycin A1-treated/-untreated targeted cells under the presence and absence of rapalog on the same blot (Fig. 6f).

5. Introduction. In line 93-95, the authors emphasized that class III PI3K-kinases complex is the hub of ubiquitin-dependent regulation. However, class III PI3K complex subunits are regulated or modified by not only ubiquitin but also many other post-translational modifications such as phosphorylation, acetylation, or SUMOylation.

6. In Figure 2C, the data interpretation is somewhat biased. Under K29/K48-only ubiquitin overexpression, TRABID-induced VPS34 deubiquitination appears to be not significantly more efficient as compared with the wild type control. Quantitation or an additional experiment is needed.

Responses to reviewers

Reviewer #1: (Remarks to the Author):

In this manuscript, authors investigated the role of E3 ligase UBE3C and deubiquitinase TRABID in the regulation of autophagy, proteostasis and liver lipid metabolism. They provided evidence to show that TRABID deubiquitinate VPS34 K29/K48 to prevent its proteasomal degradation and increased stabilization of VPS34 to promote autophagosome formation. In contrast, the E3 ligase UBE3C promotes the branched ubiquitination and proteasomal degradation of VPS34 to inhibit autophagy. They also showed physiological relevance of this pathway in the context of diet-induced non-alcoholic fatty liver disease (NAFLD). The results overall identified a novel ubiquitination regulation of VPS34 stability in response to autophagy stimuli through the reciprocally regulation by TRABID and UBE3C, which may be critical in maintaining a balanced autophagy activity. Most data are of good quality and manuscript was well written. However, some issues have to be addressed (details listed below) to improve the quality of this manuscript.

Specific comments:

1. The citation on the cross-talk of UPS and autophagy was mainly recent review papers (ref 20-23). Two pioneer original research papers on the cross-talk of ER stress, UPS and autophagy should be cited (PMID: 16901900 and PMID: 17620365).

We included the second paper in the Introduction to strengthen the link between UPS and autophagy (Line 86; Ref 23). We also cited the first paper, which reported the induction of autophagy by ER stress, to an appropriate place in the manuscript (Line 310; Ref 45).

2. Figure 1c & e, to increased data rigor and reproducibility, the experiments should be repeated for at least 3 times and densitometry analysis data should reflect these repeated experiments as these are the key fundamental data for the main conclusion of this manuscript. Similar requirements should also be applied for Figure 4.

We agree with the reviewer that these LC3 lipidation data are fundamental for the main conclusion. In the revised manuscript, all experiments in Fig. 1 and Fig. 4 involving LC3-lipidation, including Fig. 1b, 1c, 1e, 4b, 4d, and 4g, have been repeated for 3 times and the densitometry analysis data are shown.

3. Data presentation for RFP-GFP-LC3 puncta assay in Figure 2 m & n was questionable. VPS34 affects the upstream autophagosomes formation/biogenesis

(also supported by Figure 1 in this manuscript although it may also regulate autophagosome maturation), then the ratio of red vs total of LC3 puncta (from RFP-GFP-LC3 assay) would be misleading as the possible altered green only puncta numbers by manipulating TRABID. Authors should present the red and yellow puncta numbers separately in addition to the ratio data.

As requested by the reviewer, we presented the yellow and red puncta numbers, along with the ratios of red/total puncta in Fig. 2n, o (Fig. 2m, n of the original manuscript). Notably, the yellow puncta numbers are indeed decreased by TRABID knockdown and increased by TRABID overexpression.

4. Figure 3m-o, the labeling shUb-Ub should be clearly defined in the text or figure legend.

To facilitate an easy understanding, we changed the label of cell types as WT and K29R in revised manuscript. Doxycycline treatment of K29R cells leads to the depletion of endogenous ubiquitin and expression of K29R ubiquitin, which is explained in the text (Lines 275-278).

5. Figure 5, various ER stress inducers decreased the interaction of UBE3C with VPS34 and did not change its interaction with TRABID (supplemental Fig 5). While the authors tended to conclude that ER stress favors the stabilization of VPS34 to promote autophagy, data in Figure 5a did not show significant increase of VPS34. This would conflict the conclusion. As most experiments were performed under the overexpression conditions, it is important to show the endogenous ubiquitination and stabilization of VPS34 under the ER stress conditions.

(1) The reviewer criticized that VPS34 levels in the immunoprecipitates shown in Fig. 5a were not differed among the different treatments. This is due to two reasons. First, in the immunoprecipitation (IP) experiment, the cognate antibody usually cannot precipitate all antigens present in the cell lysate. Therefore, the difference in protein expression cannot be faithfully preserved by IP. Second, to ensure a fair comparison of protein-protein interaction by IP, it would be important that the level of immunoprecipitated antigen is kept equal so that one can simply judge the level of co-precipitated protein to deduce the level of interaction. Therefore, we have adjusted the loading of immunoprecipitates so that the levels of antigen (VPS34) are comparable across all lanes. We have included this information in the figure legend to avoid confusion.

(2) We included new data for the ubiquitination of endogenous VPS34 in response to ER stresses in Fig. 5c of revised manuscript. Our data showed that ER stresses decreased the ubiquitination of endogenous VPS34, a finding similar to the

overexpressed VPS34. As to the VPS34 stabilization by ER stress, the original data (Fig. 5d and Supplementary Fig. 5b) were shown with the endogenous VPS34.

6. Figure 6c, p62 interacts with LC3 via its LIR, the rapalog experiments should only affect the number of GFP-LC3 puncta numbers, it was unclear how it would affect the interaction of p62 with LC3 (green only LC3 puncta that was not colocalized with p62)? In addition to Figure 6d, western blot analysis for protein aggregates (ubiquitination) would be more quantitative and robust.

(1) Fig. 6c investigated the Impact of enforced targeting VPS34 to UBE3C on aggrephagy induced by proteotoxic stress. The total LC3 puncta cannot distinguish aggrephagy from bulk autophagy and therefore is not suitable for measuring aggrephagy. Instead, the colocalization of LC3 with p62 or Ub dots is often used for evaluating aggrephagy activity (*Cell* 131:1149-63, 2007; *JBC* 282:24131-45, 2007; *JBC* 283: 22847-57, 2008; *EMBO J* 37: e98358, 2018; *Mol Cell* 74: 330-346, 2019). This colocalization represents the recruitment of LC3-associated autophagosome membrane to the inclusion bodies/insoluble aggregates (marked by autophagy receptor such as p62) and therefore is a specific and established method for measuring aggrephagy (*Methods in Enzymology* 588: 245-281, 2017; Please see p264 of this chapter). Besides the recruitment of LC3, a recent study has revealed a new role of p62 in recruiting upstream autophagy regulators such as FIP200, PI3-kinase (VPS34 complex) and ATG16, which are recruited prior to LC3 and are required for aggrephagy, presumably by bringing the membrane sources to the forming autophagosome (*Mol Cell* 74: 330-346, 2019). This is the reason why VPS34 affects aggrephagy activity and the localization of p62 with LC3. In line with this study, we showed that persistent degradation of VPS34 by enforced targeting VPS34 to UBE3C leads to an impairment of aggrephagy, as evident by the decreased colocalization of p62 with LC3 (Fig. 6c). Of note, the green-only dots in this figure represent autophagosome formed through an aggrephagy-independent process (such as bulk autophagy). They are irrelevant to this assay and therefore are not counted.

(2) The reviewer rightly pointed out that WB analysis of ubiquitinated species in the insoluble compartment is another way for measuring aggrephagy. We included such data in Supplementary Fig. 6b of revised manuscript. Consistent with the impaired aggrephagy, our data showed that enforced targeting VPS34 to UBE3C increased ubiquitinated species in the insoluble fraction.

7. Figure 7, how did HFD decrease TRABID? Did the ER stress inducers in cultured cells also lead to the decreased TRABID? Was the protection from TRABID overexpression against diet-induced NAFLD solely due to increased autophagy?

Increased de novo lipogenesis is also implicated in HFD-induced NAFLD. Did TRABID also affect de novo lipogenesis?

The reviewers raised several questions related to NAFLD model. In revised manuscript, we included new data showing that HFD downregulates TRABID through the mRNA level (Supplementary Fig. 7a). Future study will aim to explore how *TRABID* mRNA is decreased under this condition. In addition, using AML12 mouse hepatocyte model, we showed that TRABID is downregulated by treatment of cells with oleic acid but not by tunicamycin (Supplementary Fig. 7b). Thus, it is the accumulation of free fatty acid rather than the induction of ER stress that results in the downregulation of TRABID. Furthermore, we showed that VPS34 inhibitor SAR405 greatly reduced the protection effects of TRABID on NAFLD, as evident by the reversed effects on liver weight/body weight, hepatic lipid contents and serum AST/ALT levels (Fig. 8a-e and Supplementary Fig. 7d). These data provide a substantial evidence that elevation of autophagy activity through VPS34 stabilization contributes at least in part to the protective effect of TRABID on diet-induced NAFLD. We do not know whether TRABID affects *de novo* lipogenesis. In addition, although the reviewer pointed out the contribution of an increased *de novo* lipogenesis to HFD-induced NAFLD, studies that reported no effect or even a decreased hepatic *de novo* lipogenesis in response to HFD can also be found (*Diabetes* 59: 2495-2504; *J Lipid Res* 31: 623-631, 1990; *J Lipid Res* 55: 2541-2553, 2014). Given that this manuscript focuses on the ubiquitin-dependent regulation of a key autophagy player, VPS34, together with the complex regulation of *de novo* lipogenesis by HFD, we believe that *de novo* lipogenesis is beyond the scope of this study. It would be more appropriate for a separate study.

Reviewer #2: (Remarks to the Author):

The manuscript by Chen et al. Identify TRABID and UBE3C positive and negative regulators, respectively, of autophagosome formation and maturation. This is a very comprehensive study from Chen et al. The experiments are well carried out and the conclusions are largely supported by the data. The paper would be strengthened by more experiments being done at the endogenous level but it is understood this can be very difficult to do properly and should not prevent publication.

We thank the reviewer for the support.

Comments:

The authors must tabulate the raw data from the shRNA screen presented in supplementary Figure 1a thereby revealing other proteins that potentially regulate LC3 puncta formation. This would be a valuable resource for the community.

We agree with the reviewer. The raw data are presented in Supplementary Table 1 of revised manuscript.

Authors should include a sentence explaining why experiments were carried out on both fed and starved cells.

In revised manuscript, we changed “fed and starvation” to “DMEM and EBSS” as requested by Reviewer 3. In addition, we explained that these conditions are used for measuring basal and starvation-induced autophagy activities (Lines 131-133).

I do not understand the WB in figure 1bc and e. What does the I and II annotation for the two bands mean? Presumably the knockdown efficiencies printed at the bottom of the top panels correspond to TRABID knockdown so why does the key state LC3 II? Author’s should explain how the efficiencies were calculated in the figure legend.

When autophagy is induced, LC3 protein is conjugated by phosphatidylethanoamine (PE). The unmodified and lipidated LC3 are called LC3-I and LC3-II, respectively. Thus, LC3-II level (normalized with the level of a control protein) is widely used in the autophagy field for monitoring autophagy activity (see “Guidelines for the use and interpretation of assays for monitoring autophagy in higher eukaryotes”, *Autophagy* 4: 151-175, 2008). As requested by the Reviewer 1, all experiments measuring LC3-II were repeated for three times and statistics are shown in the revised manuscript.

Supporting publications should be cited/cited again where appropriate in the results sections.

E.g. “These findings suggest an impact of TRABID on the class III PI3-kinase REFs”.

The sentence that the reviewer pointed out is based on our findings, not any previous report, since the relation of TRABID to autophagy has not been reported elsewhere. In the sentence, “these findings” mean data described in Supplementary Fig. 1b-d and Fig. 1f, g. Based on the positions of these players/markers in the autophagosome formation process, we deduced that TRABID likely acts on class III PI3-kinase, i.e., VPS34 complex. To explain this more clearly, we have modified the sentence, which also allows the citation of one reference (Lines 151-152).

Assertions are invariably derived from experiments involving a double overexpression experiment. For example, in Figure 2a it would be much more

convincing if endogenous VPS34 was being visualized rather than overexpressing both it and TRABID.

We agree with the reviewer. The ability of TRABID to reduce VPS34 ubiquitination is now validated with endogenous VPS34. The new data are included in Fig. 2b of revised manuscript. In addition, we presented data for the induction of endogenous VPS34 ubiquitination by ER stress and proteotoxic stress in Fig. 5c of revised manuscript.

Figure 2d. It is surprising that TRABID has activity towards K48 chains. However, previous specificity studies have been carried out with protein truncated to the OTU and ankrin repeat domains. The authors might want to provide an explanation for why K48 activity might be present in the context of full-length protein. E.g. Maybe the NZF domains bind K29 linkages (as shown by the Komander lab, Michel et al. 2015) and position the catalytic domain in proximity of K48 linkages.

We have provided explanations in the Discussion (Lines 530-538) for the ability of TRABID to reduce the K48 linkages on the K29/K48 branched chain conjugated to VPS34. First, when TRABID cleaves a K29 linkage, all K48 linkages located at more distal places are removed from the substrate (see figure on the right). This could explain how TRABID can reduce K48-linked ubiquitination on VPS34 in vitro (as suggested by the reviewer as a nice experiment in the following point).

In addition, we cannot rule out a possibility that a DUB capable of cleaving K48 linkage is associated with TRABID in vivo to assist the hydrolysis of K48-linkages. Regarding to the intriguing explanation provided by the reviewer, it is nevertheless inconsistent with our and previous findings. In our study, we showed in Supplementary Fig. 2e that the full-length TRABID purified from 293T cells cultured in either normal or starved condition cannot hydrolyze K48-linked di-ubiquitin. Accordingly, Komander's group reported that full-length TRABID purified from 293T cells cannot hydrolyze K48-linked di-ubiquitin (Fig. 5b in *Nat Struct Biol* 19: 62-71, 2012). Furthermore, this previous study indicated that the chain type specificity of full-length TRABID is similar to the AnkOTU fragment of TRABID (i.e., favoring K29, K33 and K63). Thus, no evidence indicates that inclusion of the three NZF domains could alter the specificity of TRABID.

A nice complementary experiment would involve preparing heterotypic K29/K48 chains in vitro using recombinant UBE3C. These could then be tested as a substrate for full-length recombinant TRABID and analyzed using the K48 antibody.

Such experiment was indeed performed and data are shown in Fig. 2h. Indeed, TRABID can reduce the K48-linkage from the polyubiquitin chain on VPS34 assembled by UBE3C.

The implementation of the ubiquitin replacement strategy is nice.

We thank the reviewer for the commendation.

Minor comments:

Some of the claims should be toned down. For example line 177 should be changed to- Collectively our findings suggest.....

We have changed it accordingly.

The following sentence like an oxymoron.

“Together, our study identifies VPS34 as a substrate of TRABID. TRABID antagonizes VPS34 K29/K48 heterotypic ubiquitination even without directly hydrolyzing the K48 ubiquitin chain.”

How about:

“Together, our study identifies VPS34 as a substrate of TRABID. TRABID antagonizes VPS34 K29/K48 heterotypic ubiquitination even though it does not hydrolyze unanchored K48 diubiquitin chains (ref.)”

We have changed the sentence and cited reference (Lines 176-178).

Line 219. Please stick to the terminology defined in the introduction and say “homotypic” rather than “linear”.

In this circumstance, we want to distinguish the two types of heterotypic chains, i.e., linear (mixed) vs. branched. We realize that the term of “linear chain” could be misinterpreted as “homotypic chain” and therefore changed it as mixed chain in revised manuscript (Lines 230 & 232).

Reviewer #3: (Remarks to the Author):

In the manuscript “VPS34 K29/K48 branched ubiquitination governed by UBE3C and TRABID regulates autophagy, proteostasis and liver metabolism” by Chen et al, the authors report that ubiquitin ligase UBE3C and deubiquitinating enzyme TRABID reciprocally target VPS34 for K29/K48 branched ubiquitination to control the protein stability of VPS34. There are several concerns (some listed below) that I

have with this study. Most of these concerns are associated with slight differences of VPS34 protein levels of most western blot images and some results are not convincing. This study is forced to put the data together and the mouse model of nonalcoholic fatty liver disease (NAFLD) is not sufficient to support the conclusion that VPS34 ubiquitination governed by UBE3C and TRABID regulates liver metabolism. Specific comments are listed below.

As described below, we have explained thoroughly regarding to each “non-convincing” data pointed out by the reviewer, have provided a set of new data to demonstrate that the effect of TRABID to alleviate NAFLD is VPS34-dependent, and have shown that TRABID downregulation in response to HFD occurs at the mRNA level and also in mouse hepatocyte cell lines stimulated with oleic acid. With these explanations and additional data, we believe that we have addressed all concerns of this reviewer.

Major comments:

1. The quality of image in Figure 1a is quite low as I cannot clearly observe the morphology of the HeLa cells. In Figure 1b-c, the authors had better to detect the expression of TRABID in the same experiment setting, thus providing the evidence that there is a positive correlation between TRABID and autophagosome formation. To further indicate that TRABID promotes autophagy process, the authors should analysis the degradation of autophagic substrate p62. In Figure 1g, we can hardly see any significant difference of the ATG16 puncta in TRABID depletion cell when compared with control samples. I strongly suggest the authors label the expression of TRABID with fluorescent antibody in the confocal microscopy experiments.

(1) For Fig. 1a, we provided new data for monitoring endogenous LC3 puncta, which are more physiologically relevant than GFP-LC3 puncta. This change also makes the methodology of Fig 1a consistent with that of all other similar experiments in the manuscript (All other experiments for single-color LC3 staining detected endogenous LC3). Furthermore, endogenous LC3 gave a weaker staining signal than GFP-LC3, so that the background (cell morphology) is more evident.

(2) Fig. 1b used stable lines carrying control or TRABID shRNAs. We originally did not provide data for the levels of TRABID expression in EBSS-starved cells as they should have no difference with those in DMEM-cultured cells. This is because EBSS starvation does not change TRABID level (Fig. 1e). However, since the reviewer requested, we included such data in revised manuscript. As expected, in both basal and starved conditions, TRABID shRNAs showed similar knockdown efficiencies and starvation did not change TRABID expression levels in control and knockdown cells.

Thus, if we compare the six lanes together, TRABID level does not completely correlate with LC3-II level. This is reasonable, because the increased LC3-II by starvation is due to the activation of AMPK-ULK1/2 axis (for review see *Mol Cell Biol* 32:2-11, 2012), rather than an upregulation of TRABID. This piece of data also indicated that TRABID positively regulates autophagy in both basal and starvation conditions, but its expression level is unaltered by starvation. Such finding is not surprising as many autophagy players (such as the key players ATG16, ATG5, ATG7....) behave similarly. In addition to providing TRABID knockdown data in all lanes, we included p62 levels in Fig. 1b. As expected, the results showed opposite regulations compared to LC3-II. Furthermore, as requested by Reviewer 1, we repeated the experiments of LC3-II and p62 for three times and statistics are shown. Collectively, these changes have provided convincing evidence for a positive role of TRABID in autophagy regulation under both basal and EBSS-starved conditions.

(3) For Fig. 1g, we replaced the endogenous ATG16 data with GFP-ATG16, as suggested by the reviewer. This change has indeed improved the quality of image and the new data are of the same conclusion as the old data. We are grateful for reviewer's suggestion.

2. The authors claimed that TRABID can cleave the K29/K48 branched ubiquitin chains on VPS34 and the ubiquitination enhances the binding of VPS34 to proteasome for degradation, however, in all the co-immunoprecipitation experiments of Figure 2, the protein levels of VPS34 are not up-regulated with TRABID overexpression in the lysate samples. The role of the K29/K48 branched ubiquitination in VPS34 stabilization should be further confirmed. It has been reported that unanchored ubiquitin chains act as critical mediators in the cellular signal transduction, the authors need check whether VPS34 is covalently attached to anchored ubiquitin chains or noncovalently with unanchored ubiquitin chains. Moreover, which lysine site of VPS34 is associated with the K29/K48 branched ubiquitination should be dissected.

(1) The reviewer criticized that VPS34 level is not upregulated in all co-IP experiments shown in Fig. 2. Notably, co-IP experiments were employed in Fig. 2a, b, c, d, e, and f. However, only Fig. 2a, b, c, d, and e involve TRABID overexpression. Therefore, we believe that the reviewer meant these five experiments. All five experiments were performed to measure VPS34 ubiquitination levels in vivo. To preserve ubiquitination, cells were treated with proteasome inhibitor MG132 for 16 h before lysis, which has been described in the first sentence of the "In vivo ubiquitination and deubiquitination assays" paragraph in the "Methods" section. Treatment of MG132 to prevent the degradation of ubiquitinated species is an important step for assaying

protein ubiquitination with a degradable fate and has been commonly used in the ubiquitin field. Thus, in the in vivo ubiquitination/deubiquitination assays, the observation of no change in the substrate level is a general phenomenon. Below, we list few recent papers that contain protein ubiquitination and/or deubiquitination assays for the reviewer to check: *Nat Cell Biol* 22: 1130-1142, 2020 (Fig. 3c); *Nat Cell Biol* 22: 1056-63, 2020 (Fig. 2c); *Nat Cell Biol* 22:1064-75, 2020 (Extended Fig. 2b); *Nat Commun* 11: 1268, 2020 (Fig. 2f); *Dev Cell* 48: 329-344, 2019 (Fig. 1I, 3F, 5D); *Dev Cell* 48: 345-360, 2019 (Fig. 1G, 2H, 2I, 4G, 5J, 5K); *Nat Commun* 9: 4648, 2018 (Fig. 1b, 1f, 1g, 6c, 6d, 7c); *Mol Cell* 71: 592-605, 2018 (Fig. 3F, 3H, 5K, 5M, 6F, 6H, 6J); *Nature* 553: 91-95, 2018 (Fig. 4d, Extended Fig. 4f, Extended Fig. 5f).

(2) We do not believe that VPS34 can bind to an unanchored ubiquitin chain as it does not contain any ubiquitin-binding domain and is not listed among the known ubiquitin-binding proteins (*Nucleic Acid Res* 46: D447-453, 2018). This notion is confirmed by the experimental data shown on the right. In this experiment, we incubated Flag-VPS34 bound on anti-Flag M2 beads with a solution containing unanchored K29/K48 branched ubiquitin chain generated by UBE3C in vitro (see our reply to point 3 of Reviewer 4 for the method and evidence of the generation of this branched chain). This pull down analysis clearly showed that VPS34 cannot bind unanchored ubiquitin chain.

(3) As requested by the reviewer, we determined the ubiquitination site on VPS34 promoted by UBE3C using LC-MS/MS and identified multiple sites. A summary of these sites are listed in Supplementary Fig. 3c of revised manuscript and the Mass Spec raw data are presented in the raw data file.

3. In the lysate images of Figure 3b, 3c, 3d, 3e and 3f, it seems that overexpression of UBE3C cannot mediate the degradation of VPS34 and knockdown UBE3C has no effect on the protein stability of VPS34. The authors should also resolve the concern that whether UBE3C degrades VPS34 in a ubiquitination modification dependent manner.

Fig. 3b is an in vitro pull down assay. For a fair comparison, we must input the same amount of VPS34 in lane 2 and lane 4 to demonstrate that the binding is specific. Fig. 3c, d, e, and f all belong to the analyses of VPS34 ubiquitination in vivo. As described above, cells were treated with proteasome inhibitor for this analysis. Thus, it is reasonable that the levels of VPS34 are comparable across the lanes.

4. Does UBE3C or TRABID affect the mRNA levels of VPS34? As VPS34 occurs in two main multiprotein complexes, referred to as complex I and complex II, which are made up of VPS34, VPS15, Beclin-1 and ATG14 (complex I) or VPS34, VPS15, Beclin-1 and UVRAG (complex II). Complex I is activated and recruited to the phagophore initiation sites mainly at the endoplasmic reticulum, to produce a pool of PtdIns3P critical for the formation and elongation of the nascent autophagosome, which encloses and isolates cytoplasmic components, whereas complex II controls endosome maturation and promotes autophagosome–late endosome/lysosome fusion. Which pool of VPS34 does UBE3C or TRABID alter?

(1) We provided new data showing that neither could TRABID nor UBE3C affect VPS34 mRNA level, which are presented in Supplementary Fig. 2f and Supplementary Fig. 4b, respectively.

(2) TRABID binds both VPS34 complex I and complex II (Supplementary Fig. 2g, h) and promotes autophagosome formation (Fig. 1a-e) and maturation (Fig. 2n, o). Likewise, UBE3C binds both complex I and complex II (Supplementary Fig. 4h) and inhibits autophagosome formation (Fig. 4a-d) and maturation (Supplementary Fig. 4i). Please also see Results section for descriptions.

5. In Figure 5c, the authors showed that 17-DMAG, 17-AGG, Puro and Tg increased VPS34 stability, but in the right panel of Figure 5a, the protein abundances of VPS34 in lysate do not change with the pharmaceutical treatments.

Please see our reply to point 5 of Reviewer 1.

6. As TRABID is a positive regulator while UBE3C is a negative regulator of autophagosome formation, why TRABID overexpression does not caused higher amounts of ATG16 puncta when compared with UBE3C-overexpressing group in Figure 5d, 5f and 5h?

The purpose of these experiments is to test the recruitment of TRABID or UBE3C to phagophores, which are marked by TagRFP-ATG16 puncta. Indeed, under the condition of equal input of TagRFP-ATG16 plasmid, ATG16 puncta number was lower in UBE3C-overexpressing cells than TRABID-overexpressing cells. In this way, one cannot be certain whether the low percentage of GFP-UBE3C puncta colocalized with TagRFP-ATG16 puncta is due to a decreased ability of UBE3C to be recruited to phagophores or a decreased number of phagophores. Thus, we slightly increased the amount of transfected TagRFP-ATG16 in the UBE3C group to make a comparable number of TagRFP-ATG16 puncta in the two groups. Such adjustment is now depicted in the figure legend.

7. It has been reported that there is a prominent diminish autophagy flux in NAFLD and several major mechanisms that may account for the impairment of autophagy in NAFLD. Downregulation of autophagy proteins (such as Beclin-1, ATG7, ATG5) (Yang et al. 2010, Cell Metabolism), decreased lysosomal acidification (Wang et al. 2018, FASEB J), defective autophagosome-lysosome fusion (Koga et al. 2010, FASEB J; Park et al. 2014, Nature Communications) may lead to decreased autophagy flux in NAFLD. Does TRABID-controlled liver metabolism to alleviate NAFLD go through VPS34? Does the decreased VPS34 level with HFD treatment associate with Beclin-1 or other ATG proteins? Does VPS34 depletion cause NAFLD? It is important to check the role of TRABID and UBE3C in NAFLD with the deficiency of VPS34.

(1) To address whether the effect of TRABID to alleviate NAFLD is mediated through VPS34, we provided a set of new data in revised manuscript showing that administration of VPS34 inhibitor SAR405 greatly reduced the protective effect of TRABID on NAFLD (Fig. 8a-e and Supplementary Fig. 7d). These findings, combined with the downregulation of TRABID and VPS34 by HFD, strongly support a role of TRABID-mediated VPS34 deubiquitination and stabilization in antagonizing NAFLD. We appreciate the reviewer for this constructive suggestion, which has further strengthened the manuscript.

(2) To address the question “whether the decreased VPS34 level with HFD is associated with beclin-1, we would like to point out that previous studies indicated that VPS34 downregulation leads to the downregulation of several other complex subunits, including beclin-1 (*Autophagy* 6: 764-776, 2008; *Exp Cell Res* 316: 3368-3378, 2010). Thus, HFD-induced TRABID and VPS34 downregulation likely results in the downregulation of beclin-1 level in the liver as well. Nevertheless, since TRABID does not promote the deubiquitination of beclin-1 or several other VPS34 complex subunits (Supplementary Fig. 2a-d), its primary effect should be on VPS34.

(3) Regarding to whether VPS34 depletion causes NAFLD, we would like to point out that this has been demonstrated previously. First, liver-specific ablation of VPS34 in mice (Albumin-Cre; VPS34^{f/f}) leads to hepatomegaly and hepatic steatosis (*PNAS* 109: 2003-2008, 2012). Furthermore, adenovirus-mediated delivery of a VPS34 mutant that is impaired in starvation-induced deacetylation/activation leads to liver steatosis (*Mol Cell* 67:907-921, 2017).

Minor comments:

The authors had better label the cells with normal cultured condition (DMEM) and starvation (EBSS), but not with Fed, as the experiments are carried out in cells but not animals.

We have changed these terms accordingly.

2. In Figure 7j, the authors need remodified their working model, as they have not evidence to demonstrate that UBE3C plays a physiologic role in mice with HFD treatment.

We agree with the reviewer and have deleted this model.

Reviewer #4: (Remarks to the Author):

VPS34 is a critical regulatory component of Beclin1 complex which controls basal and stress-dependent autophagic proteolysis. Although ubiquitin-dependent VPS34 turnover has been well-been characterized, studies have mainly focused on linkage types of ubiquitin chain and simple degradation biology with respect to homotypic ubiquitin chains. This paper written by Chen et al., elucidates mechanisms on regulating autophagic proteolysis via assembly of heterotypic ubiquitin chain on VPS34 turnover by UPS. The authors propose that TRABID and UBE3C are DUB and E3 ligase which remodel ubiquitin chains of VPS34 to generate K29/K48 branched ubiquitin chain. By addition of K29-linked ubiquitin on the basement of the K48 linked ubiquitin chain, VPS34 is targeted to the proteasome compartment with increased affinity. Overall, the major claims are supported by a set of carefully designed experiments. The results from this study provide critical information in the field of heterotypic ubiquitin chains and their mechanistic and functional analysis. Nonetheless, there are some parts that should be addressed for consideration for publication. The revised manuscript will be met with greater enthusiasm if these comments are sufficiently addressed, hopefully with some additional experiments.

We thank the reviewer for the support. As described below, we have carefully addressed each point raised by the reviewer, including providing new data.

Major comments

1. Overall, it remains unclear how the basal turnover of VPS34 is regulated. The results clearly show that VPS34 with K29/K48-linked heterotypic ubiquitin chain is degraded via the proteasome. The authors should address or at least discuss how VPS34 assembled with pre-existing K48-linked homotypic ubiquitin chain, a well-known UPS signal, is degraded. Some previous studies proposed that specific linkage types of ubiquitin chain do not affect VPS34 degradation. Any explanation

for this discrepancy?

(1) Although certain branched ubiquitin chains are formed by a sequential action of two E2 or E3 enzymes with different chain-type specificities, some HECT E3s can by itself assemble the branched chains with their intrinsic properties (*Trends Cell Biol* 29: 704-716, 2019). UBE3C, a HECT-family E3, forms K29/K48 branched, unanchored ubiquitin chain in vitro without the need of other E3 ligase (*EMBO J* 24: 4324-4333, 2005). This property is further confirmed by our study (Supplementary Fig. 3d). Additionally, we showed that UBE3C is capable of forming K29/K48 branched ubiquitin chain on VPS34 in vitro (Supplementary Fig. 3e, Fig. 3h, i). Even though we demonstrated that UBE3C can add the K29/K48 branched chain onto a K48-linked ubiquitin tetramer, as requested by this reviewer (see our reply to point 3 below), we do not aware any E3 ligase that can definitively assemble the K48 homotypic chain on VPS34. This is mainly due to the technical difficulty in characterizing the ubiquitin chain type to a complete extent, especially in the old days. To date, several ubiquitin ligases have been reported to promote VPS34 polyubiquitination. Among them, FBXL20-based Cullin1 (Cul1) E3 ligase and KLHL20-based Cullin3 (Cul3) ligase are both capable of mediating a degradable VPS34 polyubiquitination (*Gene Dev* 29: 184-196, 2015; *Mol Cell* 61: 84-97, 2016). However, the chain type produced by the former has not been characterized. For the latter, although the polyubiquitinated VPS34 reacted with a K48 chain-specific antibody (meaning the presence of K48 chain), it remains unclear whether other chain types co-exist with the K48 chain. Of note, detection by the K48 chain-specific antibody alone is insufficient to deduce the formation of a K48 “homotypic” chain, as K11/K48 branched chain (*Cell* 171: 918-933, 2017), K29/K48 branched chain (Fig. 2h of this manuscript) and perhaps other heterotypic chains containing K48-linkages are all recognized by this antibody. We consider that identification of an E3 ligase that can form the K48 “homotypic” chain on VPS34 or a full characterization of the VPS34 polyubiquitin chain type produced by Cul1-FBXL20 or Cul3-KLHL20 is beyond the scope of this study. Nevertheless, in accordance to reviewer’s comment, we added few sentences in the Discussion to point out the possibility that UBE3C may function as an E4 to extend the pre-existing ubiquitin chain on VPS34 by adding the K29/K48 branched linkages, thereby enhancing VPS34 proteasomal degradation (Lines 549-555).

(2) In contrast to these previous studies, our study does provide evidence that VPS34 undergoing K48 homotypic ubiquitination is designated for a degradation, albeit with a lower efficacy compared with the K29/48 branched chain. This is revealed in Fig. 3n. By comparing the turnover rate of VPS34 in Dox-treated K29R cells (where UBE3C can only form K48 homotypic chain on VPS34) with that in UBE3C knockdown K29R cells (where UBE3C-mediated VPS34 ubiquitination is impaired), we found that the

former is of a higher turnover rate. Thus, by using the Ub replacement cells combined with the UBE3C knockdown strategy, our study supports that VPS34 modified by K48 homotypic chain is of a degradable fate. We modified the sentences in the Results section (Lines 280-286) to more clearly explain this point.

(3) The reviewer mentioned nondegradable VPS34 ubiquitination identified previously. Indeed, Xiaochen Wang's group reported that VPS34 K63-ubiquitination in *C. elegans* by UBC-13/UEV-1 (E2) in conjunction with CHN-1 (E3) leads to VPS34 stabilization, consistent with a nondegradable fate of this chain type. This is now added in the Introduction of revised manuscript (Lines 106-108). Regarding to the VPS34 ubiquitination potentiated by the E3 enzyme malin in conjunction with the glucan phosphatase laforin (*BBA* 1867: 118613, 2020), we consider that the study is too preliminary (with unknown chain type and not well defined causal role in the functional outcome) to be included for a discussion.

2. The current study demonstrated that VPS34 regulatory mechanism is redirected to UPS upon assembly of heterotypic ubiquitin chain. The authors claim that this mechanism represents “a crosstalk of UPS-autophagy via VPS34”. In my opinion, “crosstalk” should involve the communication between UPS and autophagy to achieve proteostasis in the cell. I am not sure whether “crosstalk” can be used in this case.

We searched the manuscript thoroughly and cannot find the phrase of “a crosstalk of UPS-autophagy via VPS34”. The word of “crosstalk” or “crosstalks” has appeared twice in the manuscript, in the beginning of the fourth paragraph of Introduction and in the first sentence of Results. We have modified the two sentences so that “crosstalk” is no longer seen in the manuscript, except for the Reference section.

3. One remaining question arising from this study is the exact lysine residue of VPS34 responsible for heterotypic ubiquitin chain. Is it K419? There are some literatures which propose different E3 ligases responsible for ligating K48-linked ubiquitin chain on VPS34. It is unclear whether UBE3C responsible for assembly of both K48- and K29-linked ubiquitin chain at the identical residue or UBE3C can build K29-linked ubiquitin chain on any of the K48-linked ubiquitin chain basement.

(1) We analyzed VPS34 ubiquitination sites promoted by UBE3C in vivo using LC-MS/MS and identified multiple sites. Such data are summarized in Supplementary Fig. 3c and the Mass Spec raw data are presented in the “raw data” file.

(2) Despite the identification of multiple K residues whose ubiquitination is potentiated by UBE3C, current technologies do not allow the determination of a particular type of branched ubiquitin chain that is built on a particular lysine residue

of the substrate. This is due to the inability to keep the modified lysine residue and the polyubiquitin chain in the same peptide for Mass Spec analysis. Indeed, a recent review article has pointed out the need to develop new technologies to “sequence’ the linkages (*Trends Cell Biol* 29: 704-716, 2019, see the 1st paragraph of the “Concluding Remarks and Future Perspectives” section) and another recent review article has listed this as one of the “Outstanding Questions” in the field of branched ubiquitination (*Molecules* 25: 5200, 2020; doi:10.3390). Nevertheless, since we have shown the presence of di-GG modified ubiquitin by Ub-clipping analysis of the polyubiquitin chain built on VPS34 by UBE3C (Fig. 3h). This indicates that a least a portion of K29-linked chains and K48-linked chains are assembled on the same lysine residue of VPS34 as the form of branched chains (see figure on the right, adapted from *Nature* 572: 533-537, 2019). Notably, the K29/K48 branched chain generated by UBE3C contains a higher amount of K48 linkages than K29 linkages (*Mol Cell* 58: 95-109, 2015). This means that the K29 linkages are decorated on a K48 homotypic chain. However, due to the technical limitation for determining the exact chain architectures at a single-molecule and a single-residue level, we do not aware whether the K29 linkages are evenly distributed to the K48 chains conjugated to different lysine residues of the substrate and different molecules of the same or different substrates or whether their distribution exists a high degree of heterogeneity depending on the substrate architecture and the local environment of lysine residue. Nevertheless, given that the percentage of branched chain that UBE3C generated on VPS34 (15.7%, Fig. 3h) is similar to that on ubiquitin monomer (12.5%, Supplementary Fig. 3d), it seems to favor the former scenario. If this is the case, we argue that most of the long polyubiquitin chains (> 6-8 Ub moieties) generated by UBE3C are of a branched nature. Nevertheless, all of these are speculations. The definitive answer requires technology advances.

(3) Regarding to whether UBE3C can form a branched chain on K48-linked ubiquitin chain basement, we consider this as a central question for the function of UBE3C. Indeed, UBE3C and its yeast orthologue Hul5 have long been thought to function as the E4 enzymes on proteasome, where they elongate ubiquitin chains on proteasome-bound substrates to enhance their degradation (*Cell* 127: 1401-1413, 2006; *PNAS* 114: E3404-3413, 2017). However, such E4 function has never been

directly demonstrated. Since the reviewer inquired, we thought to test the E4 function by using the K48-linked Ub-tetramer as a substrate. As shown in the figures below, UBE3C can indeed elongate the Ub tetramer to form a longer chain (panel A). Of note, if the ubiquitin chains were entirely assembled from Ub monomer, we would have seen an increased intensity of the band corresponding to Ub₄. However, the intensity of this band is decreased upon UBE3C-catalyzed ubiquitination, indicating that at least a portion of the branched chain is assembled from the Ub₄ base. We then used Ub-clipping method combined with intact MS analysis to show that the unanchored polyubiquitin chain contains branch points (panel B). Finally, LC-MS/MS analysis of the Ub-clipping products indicated the presence of only K29- and K48-linkages (panel C). Thus, this set of data demonstrated that UBE3C can assemble K29/K48 branched chain on a pre-existing K48-linked homotypic chain and therefore validated its E4 function. Nevertheless, we feel that inclusion of this set of data would disrupt the flow of our manuscript, as the manuscript focuses on VPS34 branched ubiquitination, not the function of UBE3C. Thus, we have decided to save these data for a separate and, perhaps, UBE3C-centered study.

4. The authors proposed that both ubiquitin-dependent and -independent ER-phagy under ER stress are regulated by UBE3C and VPS34 (Fig. 6e). However, a previous study identified the ER transmembrane receptor protein TRIM13 as a K63-linked

ubiquitination substrate which recruits autophagic adaptor p62 for ubiquitin mediated ER-phagy. Given that the ER is a site of omegasome generation, co-localization assay of LC3 with the membrane protein calnexin is not sufficient to claim that the UBE3C-VPS34 pathway mediates ubiquitin-dependent selective ER-phagy. The author may need to clarify this point by assessing the role of UBE3C-VPS34 pathway on the autophagic co-localization and turnover of ER-resident luminal proteins. Also, autophagic flux assay should be correctly performed by comparing bafilomycin A1-treated/-untreated targeted cells under the presence and absence of rapalog on the same blot (Fig. 6f).

(1) The TRIM13 paper has indeed been cited by us as the reference for ubiquitin-dependent ER-phagy. In fact, the TRIM13 study not only has no conflict with our study but actually supports an important role of VPS34 complex in TRIM13-mediated ER-phagy. In this paper (*Mol Cell* 75: 1058-1072, 2019), the authors demonstrated the interactions of TRIM13 with VPS34 and Beclin-1, which are further strengthened by ER stress (Fig. 4M). Furthermore, in response to ER stress, TRIM13 forms puncta which colocalize with WIPI2 puncta (Fig. 4N). Based on these and other data, the authors suggest that “oligomeric TRIM13 forms a platform on or near the omegasome for VPS34 complex and p62 for eventual engulfment by the isolation membrane”. Consistent with an important role of VPS34 in this type of ER-phagy, our study indicated that ER stress suppresses the UBE3C-induced VPS34 downregulation to promote ER-phagy”. Thus, ER stress can elicit more than one mechanism to induce TRIM13-mediated ER-phagy, which, we believe, is reasonable.

(2) We greatly appreciate the reviewer for rightly pointing out the methodology problem for monitoring ER-phagy. In revised manuscript, we changed the methodology completely by using the ER-phagy flux reporter ssRFP-GFP-KDEL, a legitimate and general reporter for monitoring ER-phagy flux (*Trends Cell Biol* 30: 384-398, 2020). We found that enforced targeting of UBE3C to VPS34 decreased ER-phagy flux, as evident by a reduction of red puncta from confocal analysis (Fig. 6e) and a decrease in the cleavage of reporter construct from Western blot analysis (Fig. 6f). Importantly, the later was blocked by bafilomycin A1 treatment (Fig. 6f), indicating a lysosome-dependent cleavage.

5. Introduction. In line 93-95, the authors emphasized that class III PI3K-kinases complex is the hub of ubiquitin-dependent regulation. However, class III PI3K complex subunits are regulated or modified by not only ubiquitin but also many other post-translational modifications such as phosphorylation, acetylation, or SUMOylation.

Indeed, VPS34 complex components have been found to be regulated by many types

of PTMs. Although this study focuses on ubiquitination, we briefly mentioned other PTMs in the Introduction of revised manuscript (Lines 94-95) to accommodate the comment of reviewer.

6. In Figure 2C, the data interpretation is somewhat biased. Under K29/K48-only ubiquitin overexpression, TRABID-induced VPS34 deubiquitination appears to be not significantly more efficient as compared with the wild type control. Quantitation or an additional experiment is needed.

We tuned down the statement by changing the sentence to “The K29/K48-only ubiquitin could confer TRABID-induced VPS34 deubiquitination as the wild type ubiquitin” (Lines 160-162).

REVIEWERS' COMMENTS

Reviewer #1 (Remarks to the Author):

Authors have done an excellent job and provided additional experimental data in the revised manuscript. This reviewer is satisfied for the revision.

Reviewer #2 (Remarks to the Author):

The authors have addressed all of my concerns and I recommend publication.

Reviewer #3 (Remarks to the Author):

All the questions have been addressed.

Reviewer #4 (Remarks to the Author):

All my major comments were sufficiently addressed with an abundant set of additional data, including those regarding branched ubiquitin chains and ER-phagy flux assays. The manuscript should be accepted for publication as it is.

Point-by-point response to reviewers' comment

REVIEWERS' COMMENTS

Reviewer #1 (Remarks to the Author):

Authors have done an excellent job and provided additional experimental data in the revised manuscript. This reviewer is satisfied for the revision.

We thank the reviewer for the support.

Reviewer #2 (Remarks to the Author):

The authors have addressed all of my concerns and I recommend publication.

We thank the reviewer for the support.

Reviewer #3 (Remarks to the Author):

All the questions have been addressed.

We thank the reviewer for the support.

Reviewer #4 (Remarks to the Author):

All my major comments were sufficiently addressed with an abundant set of additional data, including those regarding branched ubiquitin chains and ER-phagy flux assays. The manuscript should be accepted for publication as it is.

We thank the reviewer for the support.